# ASH1L-MRG15 methyltransferase deposits H3K4me3 and FACT for damage verification in nucleotide excision repair

Corina Maritz [1,3], Reihaneh Khaleghi [1,3], Michelle N. Yancoskie [1,3], Sarah Diethelm [1], Sonja Brülisauer[1], Natalia Santos Ferreira[1], Yang Jiang [2], Shana J. Sturla[2] & Hanspeter Naegeli [1] ✉

To recognize DNA adducts, nucleotide excision repair (NER) deploys the XPC sensor, which detects damage-induced helical distortions, followed by engagement of TFIIH for lesion verification. Accessory players ensure that this factor handover takes place in chromatin where DNA is tightly wrapped around histones. Here, we describe how the histone methyltransferase ASH1L, once activated by MRG15, helps XPC and TFIIH to navigate through chromatin and induce global-genome NER hotspots. Upon UV irradiation, ASH1L adds H3K4me3 all over the genome (except in active gene promoters), thus priming chromatin for XPC relocations from native to damaged DNA. The ASH1L-MRG15 complex further recruits the histone chaperone FACT to DNA lesions. In the absence of ASH1L, MRG15 or FACT, XPC is misplaced and persists on damaged DNA without being able to deliver the lesions to TFIIH. We conclude that ASH1L-MRG15 makes damage verifiable by the NER machinery through the sequential deposition of H3K4me3 and FACT.

The DNA double helix is compacted at multiple layers to accommodate diploid genomes into the narrow space of a cellular nucleus[1]. As DNA is inexorably attacked by chemical and physical agents[2], this genome compaction remains adjustable to allow for DNA repair to occur at the right time and location. The need for repair factors to navigate through chromatin is illustrated by the global-genome nucleotide excision repair (GG-NER) reaction[3,4], because this system removes DNA lesions from the entire genome. Among other adducts, GG-NER excises mutagenic cyclobutane pyrimidine dimers (CPDs), which are cross-links between adjacent bases induced by the ultraviolet (UV) radiation of sunlight[5,6]. CPDs arise uniformly across chromosomes, including in DNA associated with the histones of nucleosomes that constitute the building unit of chromatin[7,8]. The relevance of a genome-wide excision of CPDs is demonstrated by the extreme solar hypersensitivity and skin cancer risk resulting from GG-NER defects in xeroderma pigmentosum (XP) patients[9,10].

To stimulate the excision of CPDs, a lesion receptor known as DDB2 (Damaged DNA-Binding 2), together with the DDB1-cullin 4A ubiquitin ligase, mediates the recruitment of an initiator complex consisting of XPC, RAD23B and centrin 2[11–14]. As XPC detects DNA damage indirectly by sensing destabilized base pairs[15,16], lesion recognition is not completed until the 7-subunit core TFIIH (Transcription Factor IIH) complex is recruited for the scanning of damaged strands by its XPD helicase subunit[17,18]. This search process culminates in DNA damage verification if the XPD helicase (as part of the TFIIH complex) remains sequestered at a lesion[19]. Once immobilized, TFIIH aided by XPA and RPA (Replication Protein A) forms an unwound DNA intermediate, in which the damaged strand is cleaved by XPG and XPF-ERCC1 (Excision Repair Cross-Complementing 1)[20,21]. This dual incision allows for the release of an oligonucleotide of ~30 residues containing the offending damage[22,23]. Finally, DNA integrity is restored by the synthesis and ligation of repair patches[24–26].

We reported that DDB2, in addition to associating with the XPC and DDB1-cullin 4A complexes, also recruits the histone methyltransferase ASH1L (Absent, Small, or Homeotic discs 1-Like)[27]. ASH1L was discovered as a member of trithorax transcriptional regulators

[1]Institute of Pharmacology and Toxicology, University of Zurich-Vetsuisse, Zurich, Switzerland. [2]Department of Health Sciences and Technology, ETH Zurich, Zurich, Switzerland. [3]These authors contributed equally: Corina Maritz, Reihaneh Khaleghi, Michelle N. Yancoskie. ✉e-mail: hanspeter.naegeli@uzh.ch

essential for normal development[28,29], adult organ function and fertility[30]. This lysine methyltransferase is expressed in many tissues including the skin[31–33]. When ASH1L was depleted via small interfering RNA (siRNA), the GG-NER response to UV lesions was diminished[27]. However, considering the large size of ASH1L (333 kDa) and its numerous interaction domains[34], it was not clear whether this chromatin modifier participates in repair as a structural scaffold or whether its methyltransferase activity modifies chromatin for the assembly of GG-NER factors. If the latter, further questions arise as to which histone residue is targeted by ASH1L to promote repair.

Here, we elucidate the molecular mechanism by which ASH1L stimulates GG-NER activity. During the first 6–12 h following a UV irradiation pulse, the GG-NER machinery is nearly unable to excise CPDs in the absence of ASH1L or its regulatory cofactor MRG15 (MORF4-Related Gene on chromosome 15). Rescue experiments revealed that this repair function of ASH1L is entirely mediated by its carboxy-terminal domain (CTD) endowed with methyltransferase activity. Genome-wide tracks of histone methylation, CTD and XPC occupancy as well as CPD excision establish H3K4me3 as the histone methylation product, whose buildup in chromatin delineates sites for XPC relocations in response to DNA damage. The ASH1L-MRG15 complex additionally recruits the histone chaperone FACT (FAcilitates Chromatin Transcription) for the follow-up delivery of DNA lesions to the TFIIH verifier. That XPC is coupled to TFIIH through the deposition of H3K4me3 and FACT, which are typically enriched at transcriptional start sites, implies that the ASH1L-MRG15 complex enables damage verification by decorating DNA lesions with epigenetic attributes of active gene promoters.

## Results

### ASH1L stimulates GG-NER activity
The human *ASH1L* gene contains 28 exons spread over 228 kilobases. To abolish its expression in U2OS cells, CRISPR/Cas9 was directed to exons 2 and 11/12 of the *ASH1L* gene using appropriate guide RNAs (Supplementary Fig. 1). This generated a 61-base pair (bp) deletion in exon 2 and a 310-bp deletion across exons 11 and 12 (Fig. 1a). Immunoblot analyses of cell lysates demonstrated the absence of ASH1L protein in ASH1L$^{-/-}$ cells, whereas the expression of core NER factors remained unaffected (Fig. 1b).

Compared to ASH1L-proficient (wild-type) controls, the *ASH1L* deletion caused a sensitization to UV light, which manifested as impaired colony formation, a reduced population of S-phase cells and an increased fraction of cells in G2/M (Supplementary Fig. 2). This ASH1L deletion also resulted in a strong DNA repair defect. Cells were UV-irradiated and allowed to recover to compare the kinetics of CPD excision (Fig. 1c). After 6 h of repair, ~30% of CPDs were removed in ASH1L-proficient cells, but essentially no CPD excision was detected in ASH1L$^{-/-}$ cells. Following 24 h of repair, ~50% of CPDs were removed in ASH1L-proficient cells but CPD excision during this 24-h period amounted to ~25% in the ASH1L$^{-/-}$ background. These time courses highlight that ASH1L is needed for an initial burst of CPD excision during the first 6–12 h following DNA damage induction.

The requirement of ASH1L for NER activity was substantiated by in situ immunofluorescence assays. U2OS cells were UV-irradiated through 5-μm filter pores to generate nuclear spots of damage (Fig. 1d). A 3-h recovery was allowed for the repair of pyrimidine-pyrimidone (6–4) photoproducts, which constitute a minor fraction of UV lesions that is rapidly cleared with a half-life of ~1 h[27]. Thereafter, the culture medium was supplemented for 1 h with the nucleoside analog 5-ethynyl-2′-deoxyuridine (EdU). Because the NER pathway removes DNA adducts including CPDs as part of oligonucleotides, which are replaced by repair patches, the incorporation of EdU in UV spots, also denoted as unscheduled DNA synthesis (UDS), is a direct measure of repair activity. This assay confirmed that the levels of CPD

excision were drastically reduced in ASH1L$^{-/-}$ cells compared to ASH1L-proficient controls (Fig. 1e–g).

The low repair activity in ASH1L$^{-/-}$ cells is comparable to that of XPC$^{-/-}$ cells defective in GG-NER activity (Fig. 1g). This residual repair was, however, higher in ASH1L$^{-/-}$ than in XPA$^{-/-}$ or XPF$^{-/-}$ cells, which are defective in both GG-NER and the transcription-coupled branch known as TC-NER[35,36]. As GG-NER excises 90–95% of all DNA adducts and TC-NER only the remaining 5–10%[37], the severe DNA repair reduction observed upon *ASH1L* deletion is consistent with an involvement of this methyltransferase in the GG-NER reaction. The additional depletion of CSB (Cockayne Syndrome B), a TC-NER factor, further reduced UDS in ASH1L$^{-/-}$ cells (Fig. 1h; see Supplementary Fig. 3 for the efficiency of siRNA-mediated depletions), thus establishing that ASH1L is required for GG-NER activity in response to CPDs in the same manner as XPC, and that the remaining excision in ASH1L$^{-/-}$ cells is mostly due to the TC-NER pathway.

### The methyltransferase function of ASH1L is essential for GG-NER stimulation
To understand how ASH1L participates in GG-NER activity, we performed rescue experiments by transfection of ASH1L$^{-/-}$ cells with a vector that codes for a 110 kDa carboxy-terminal domain (CTD) of ASH1L comprising the catalytic SET [for Su(var)3–9, Enhancer-of-zeste and Trithorax] subdomain and all known protein interaction sites (Fig. 2a). No function has yet been assigned to the remaining large amino-terminal part (~220 kDa), except that it contains putative DNA-binding motifs designated "AT-hooks"[34]. Previously, it was shown that an alanine substitution at the conserved position 2260 (F2260A) within the CTD region abolishes the histone methyltransferase activity[38]. Consequently, we also generated a vector coding for a CTD fragment carrying this F2260A change, henceforth designated CTD$_{inactive}$. Both CTD and CTD$_{inactive}$ were expressed as hemagglutinin (HA)- (Fig. 2b) and Flag-tagged fusion proteins.

Next, ASH1L$^{-/-}$ cells were transfected with empty vector (EV) or with vectors coding for CTD or CTD$_{inactive}$. After 24 h, the transfected cells were UV-irradiated through filters with 5-μm pores to generate nuclear spots of damage as outlined in Fig. 1d. After another 3 h, the cells were supplemented for 1 h with EdU followed by the analysis of EdU incorporation within UV spots. This assay revealed that the expression of CTD, but not the expression of CTD$_{inactive}$ rescued the GG-NER defect of ASH1L$^{-/-}$ cells (Fig. 2c–e). The CTD fragment restored GG-NER activity in ASH1L$^{-/-}$ cells to a level indistinguishable from that of ASH1L-proficient counterparts (Fig. 2e). Instead, CTD$_{inactive}$ failed to complement repair activity, indicating that the methyltransferase function of ASH1L is essential for its role in the GG-NER pathway.

### GG-NER stimulation depends on MRG15 but not on MLL1
The enzymatic SET subdomain of ASH1L is flanked by a regulatory module, which protrudes a self-inhibitory loop that covers the catalytic site and blocks its methyltransferase activity[38]. ASH1L is activated when this inhibitory loop is released from the catalytic site upon interaction with MRG15[39–42]. This regulatory mechanism was exploited to suppress ASH1L enzymatic activity by downregulation of MRG15. Wild-type cells were transfected with siRNA against MRG15 (or with non-coding RNA) and, 48 h later, UV-irradiated through 5-μm filter pores to generate lesion spots as in Fig. 1d. The measurement of UDS in these UV spots revealed that, like ASH1L itself, MRG15 is required for the GG-NER stimulation (Fig. 2f–h). The quantification of UDS over several experiments confirmed that the depletion of MRG15 (by two distinct siRNAs) diminishes GG-NER activity at 3–4 h after irradiation as much as did the absence of ASH1L (Fig. 2h). These results confirmed that ASH1L is involved in the GG-NER pathway through its methyltransferase function.

The recruitment of MRG15 for ASH1L activation occurs in a damage-dependent manner, as revealed by transfection of U2OS cells

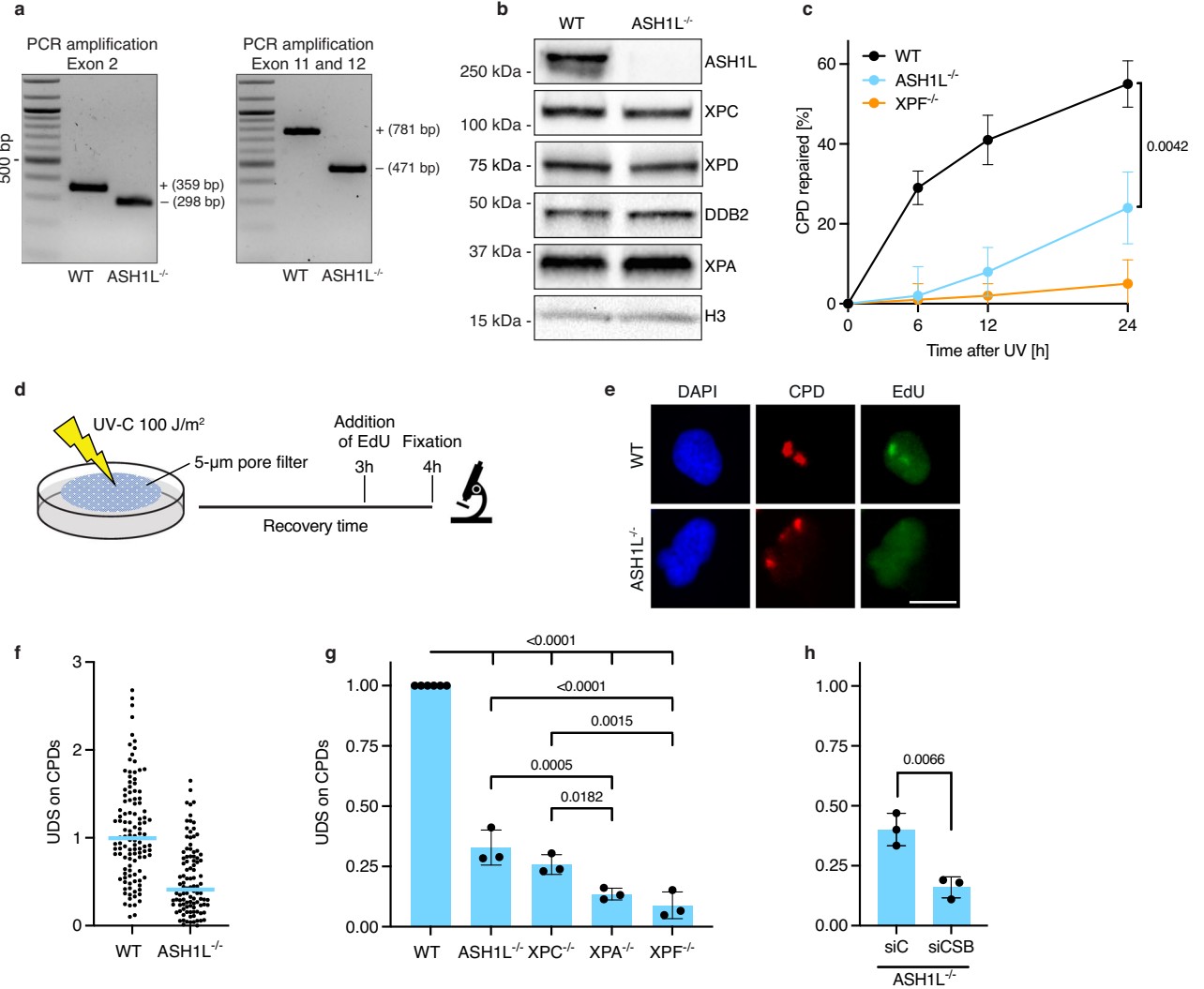

**Fig. 1 | ASH1L stimulates the GG-NER reaction.** Deletion of the *ASH1L* gene inhibits GG-NER activity during the first 6–12 h after a UV radiation pulse. **a** PCR amplifications demonstrating homozygous deletions of 61 and 310 bp in exon 2 and exons 11–12, respectively, of the *ASH1L* gene; WT, wild-type. **b** Immunoblot demonstrating the absence of ASH1L protein in ASH1L−/− cells. Whole-cell lysates were probed using antibodies against the indicated proteins. **c** Excision of CPDs in ASH1L-proficient (WT), ASH1L−/− and XPF−/− cells exposed to UV-C radiation (10 J m−2). CPDs were quantified at the indicated time points by an enzyme-linked immunoassay. Mean values ± SD, n = 3 independent experiments with three repli-cates each. The indicated *P* value for significance was determined by one-way analysis of variance (ANOVA). **d** Scheme illustrating unscheduled DNA synthesis (UDS) measurements by EdU incorporation. Nuclear spots containing CPDs were generated by UV-C irradiation (100 J m−2) through 5-μm filter pores, followed by a 3-h recovery. **e** Immunofluorescence images illustrating EdU incorporation in CPD spots of WT but not ASH1L−/− cells. DNA was stained with 4′,6-diamidino-2-phe-nylindole (DAPI). Scale bar, 15 μm. **f** Quantification of a UDS experiment based on the analysis of 100 nuclear CPD spots per condition. Horizontal lines show mean EdU incorporations in WT and ASH1L−/− cells. **g** Quantification of three independent UDS assays with WT and gene-deleted cells (from ASH1L−/− to XPF−/−). Mean values ± SD; one-way ANOVA. **h** Cockayne syndrome B (CSB) was depleted in ASH1L−/− cells by treatment with siRNA (siC, control siRNA; n = 3 independent experiments, mean values ± SD). Significance between UDS levels was determined by the two-tailed *t* test. For the experiments of (**a**, **b**), two independent replicates were conducted obtaining similar results.

with constructs coding for wild-type or mutant MRG15 conjugated with the Flag tag (Supplementary Fig. 4a). Lysates from transfected cells expressing Flag-MRG15 were subjected to pull-downs using anti-Flag antibodies. The immunoprecipitated proteins were eluted and identified by mass spectrometry (Supplementary Fig. 4b) as well as immunoblotting (Supplementary Fig. 4c). These pulldowns revealed that MRG15 interacts with DDB1-DDB2, thus indicating that, in addition to facilitating the recruitment of XPC[43] and ASH1L[27], the DDB1-DDB2-cullin 4 A complex also attracts MRG15 to activate the methyl-transferase function of ASH1L at UV lesions.

We likewise tested the contribution of MLL1 (Mixed Lineage Leukemia 1) because ASH1L has been reported to cooperate with MLL1 to generate histone marks[44] that stimulate NER activity[45]. At 3–4 h after UV irradiation, however, a depletion of MLL1 did not suppress GG-NER

activity as did the ASH1L depletion (Fig. 2i–k), indicating that the two histone methyltransferases function differently in the UV damage response.

## ASH1L reshapes the H3K4me3 landscape upon DNA damage

XPC has been shown by chromatin immunoprecipitation (ChIP) to localize, in the absence of DNA damage, to active gene promoters characterized by the enrichment of H3K4me3 histone marks[46]. Having established that the GG-NER stimulation by ASH1L relies on its methyltransferase activity, this known link between XPC and H3K4me3 prompted us to monitor the impact of ASH1L on genome-wide H3K4me3 tracks by conducting ChIP-sequencing (ChIP-seq) assays using anti-H3K4me3 antibodies. ChIP-seq reads obtained from wild-type U2OS cells were aligned to the hg38 reference genome and

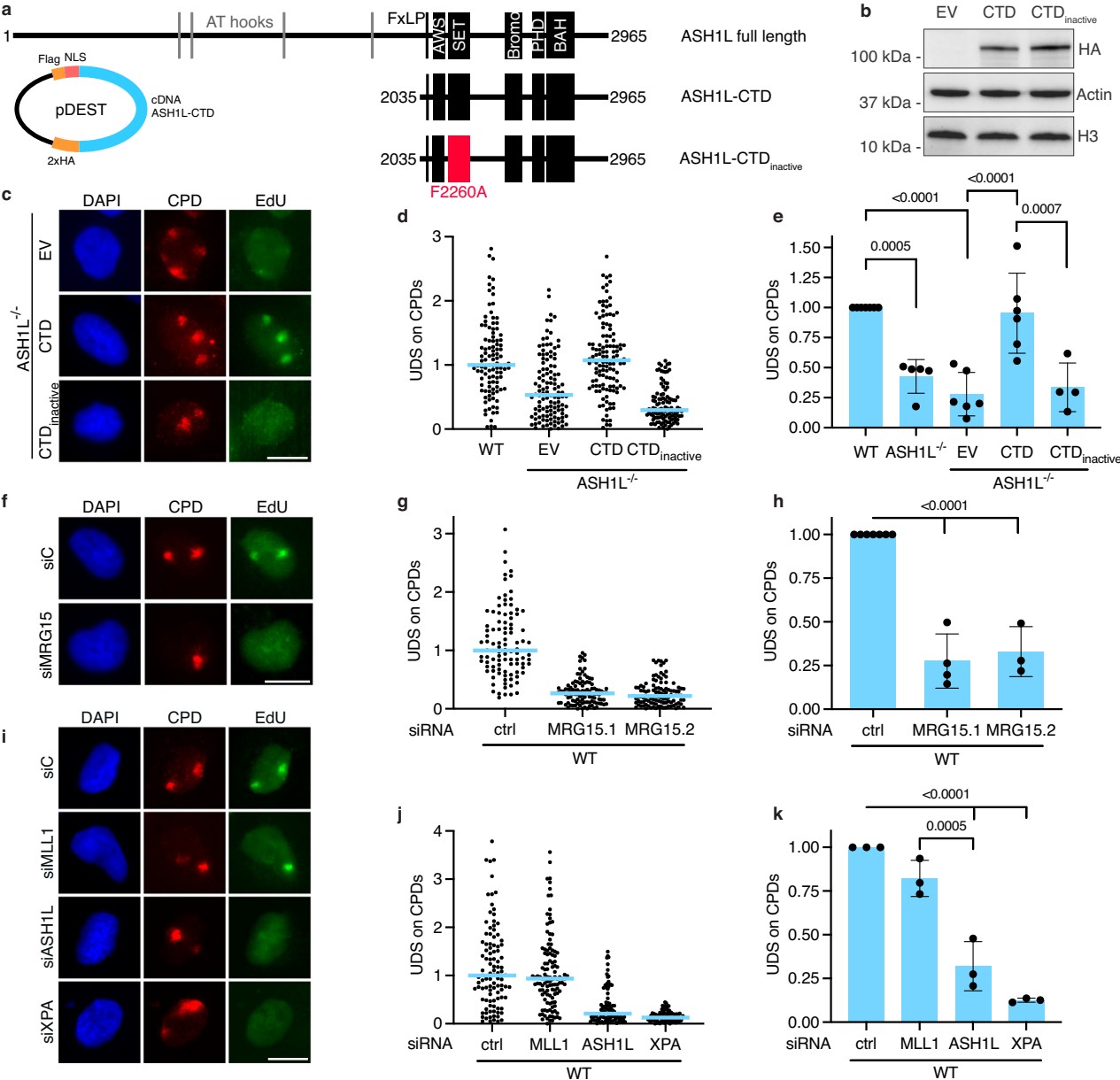

**Fig. 2 | ASH1L uses its methyltransferase function to stimulate GG-NER activity.**
An enzymatically active carboxy-terminal domain (CTD) rescues GG-NER activity.
**a** Structure of full-length ASH1L and CTD fragments. The cDNA for CTD was cloned
into vector pDEST. FxLP MRG15-interacting motif, AWS associated with SET, SET
catalytic subdomain, PHD plant homeodomain finger (a reader of methylated histone H3), BAH bromo-adjacent homology. The F2260A change abolishes methyltransferase activity[38]. **b** Immunoblot of cell lysates demonstrating the expression of
hemagglutinin (HA)-tagged CTD and CTD$_{inactive}$ in ASH1L$^{-/-}$ cells. Actin and histone
H3 were internal standards. Two independent replicates were conducted obtaining
similar results. **c** Immunofluorescence images illustrating the rescue of UDS by
transfection of ASH1L$^{-/-}$ cells with vector coding for CTD, but not CTD$_{inactive}$ or
empty vector (EV). **d** Quantification of a UDS assay by the analysis of 100 CPD spots
per condition. Horizontal lines show mean EdU incorporations under the indicated
conditions. **e** UDS assays with WT and ASH1L$^{-/-}$ cells, the latter transfected with

vectors for CTD and CTD$_{inactive}$ expression. Mean values ± SD ($n = 4$, 5 or 6 independent experiments, as indicated); $P$ values were determined by one-way ANOVA.
**f** Immunofluorescence images illustrating the UDS inhibition in WT cells upon
MRG15 depletion. **g** Quantification of a UDS assay by the analysis of 100 CPD spots.
Two different siRNA sequences were used for MRG15 depletion (ctrl, U2OS cells
transfected with control siRNA). **h** Quantification of UDS assays in WT cells treated
with the indicated siRNA sequences. Mean values ± SD ($n = 4$ or 5 independent
experiments, as indicated); one-way ANOVA. **i** Immunofluorescence images illustrating that UDS remains efficient upon MLL1 depletion. **j** Quantification of a UDS
assay comparing ASH1L and MLL1 depletions by the analysis of 100 CPD spots.
**k** Quantification of three independent UDS assays comparing ASH1L and MLL1
depletions (mean values ± SD); one-way ANOVA. Scale bars in (**c**, **f**, **i**) correspond
to 15 μm.

mapped to genomic features (see Supplementary Fig. 5a for the
baseline H3K4me3 distribution). Distinctive peaks of histone methylation became evident after filtering the H3K4me3-associated
sequences for read pileups. In unchallenged cells, these H3K4me3
peaks were detected throughout the genome but enriched as expected
in the promoters of active genes. Gene bodies, intergenic regions and

the promoters of inactive genes contained substantially fewer
H3K4me3 peaks than did active promoters (Fig. 3a).

Next, a UV dose (20 J m$^{-2}$) estimated to generate ~1 CPD per 10
kilobase pairs[47] (~1 CPD per 50 nucleosomes) was applied, after which
the cells were allowed to recover for 1 or 3 h. While the overall
H3K4me3-ChIP-seq signal decreased in both ASH1L-proficient and

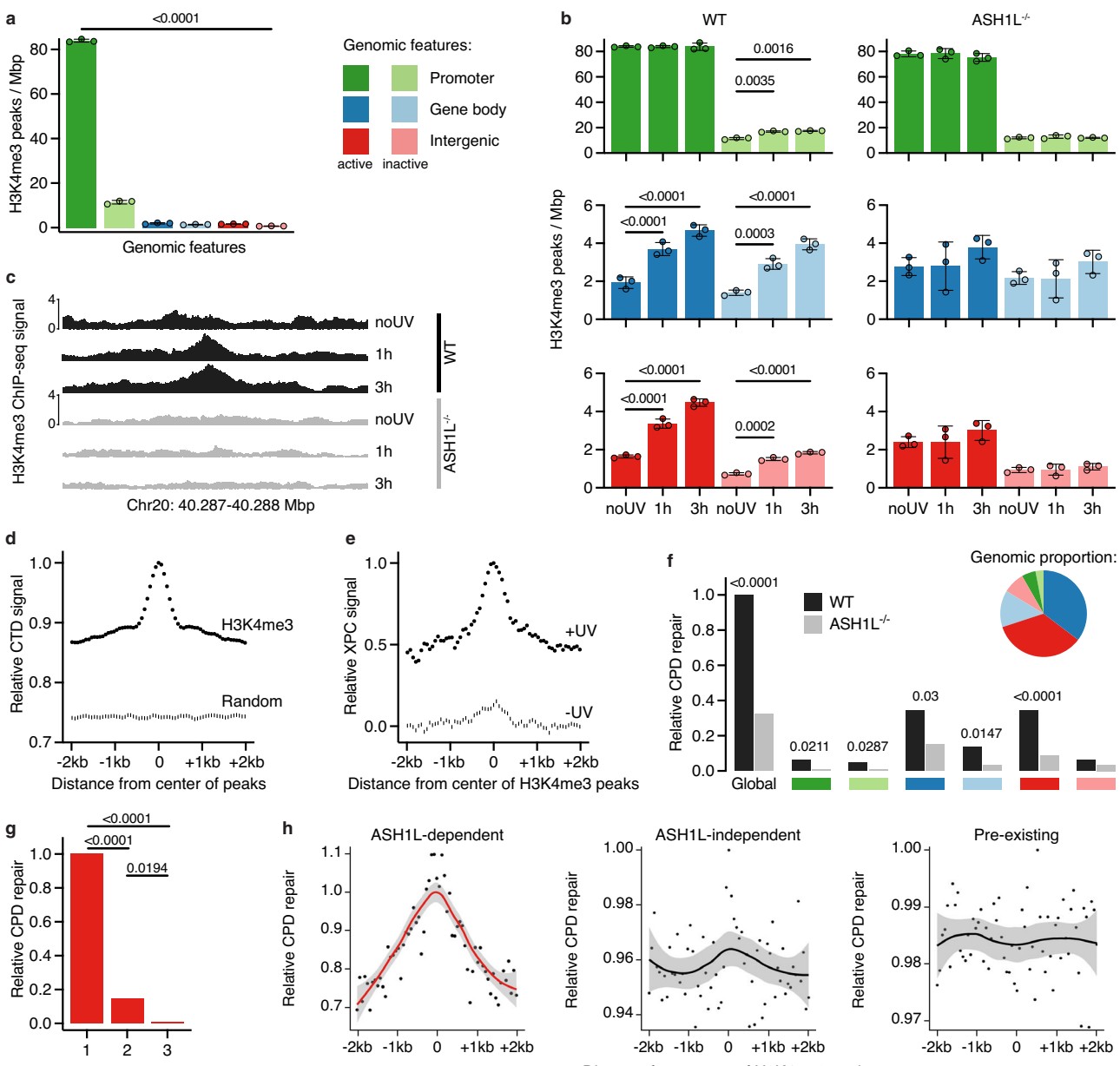

**Fig. 3 | ASH1L stimulates GG-NER by H3K4me3 deposition.** Genome-wide alignments reveal that histone methylation by ASH1L drives XPC relocations to induce GG-NER. **a** Baseline H3K4me3 peak densities in WT cells. H3K4me3-ChIP-seq reads were filtered for peak height with 75th percentiles as threshold (mean values of three independent experiments ± SD). The genome was dissected as indicated by colors, active and inactive intergenic regions are euchromatin and heterochromatin, respectively. *P* values were determined by one-way ANOVA. **b** Changes of H3K4me3 peak density 1 and 3 h post UV (20 J m$^{-2}$) in WT and ASH1L$^{-/-}$ cells. Same color code as in (**a**). Mean values ± SD ($n = 3$ independent experiments, one-way ANOVA). **c** H3K4me3-ChIP-seq tracks spanning 1000 bp of chromosome 20, illustrating a novel ASH1L-deposited H3K4me3 peak. **d** CTD occupancy after UV irradiation. Shown are CTD-ChIP-seq reads in 64-bp bins around the centers of novel ASH1L-deposited H3K4me3 peaks (top line) and in random genomic regions (bottom line). Mean values of three independent CTD-ChIP experiments. **e** XPC occupancy in WT cells. Shown are XPC-ChIP-seq reads, with background from

ASH1L-deficient cells (Supplementary Fig. 5b), in wild-type cells read pileups generating H3K4me3 peaks increased across the genome upon UV exposure (Fig. 3b, left row of bar plots). In euchromatic intergenic regions, for example, the UV challenge induced a threefold increase in H3K4me3 peak density. Within the first 3 h after irradiation, ~55,000

ASH1L$^{-/-}$ cells subtracted, at sites of novel ASH1L-deposited H3K4me3 peaks, before (bottom line) and 1 h after UV treatment (top line). Mean values of four independent XPC-ChIP experiments. **f** CPD excision over 3 h post UV determined by HS damage-seq covering the genome overall and separate genomic features. Black, WT; grey, ASH1L$^{-/-}$ cells; pie chart, DNA fraction in each feature. **g** CPD excision over 3 h in three sub-features of intergenic regions in WT cells: 1, novel ASH1L-deposited H3K4me3 peaks; 2, novel ASH1L-independent H3K4me3 peaks; 3, random regions. CPD excision is normalized for the varying number of 654-bp bins in each sub-feature. **f**, **g** Two-tailed Wilcoxon rank-sum tests were applied. **h** Correlation between H3K4me3 and CPD excision in WT cells during 3 h. Shown are mean excision values from 64-bp bins centered around novel ASH1L-deposited H3K4me3 peaks in intergenic regions (left). Genome-wide peaks are included for comparison: H3K4me3 peaks formed post UV independently of ASH1L (center) and preexisting H3K4me3 peaks (right). Shaded areas represent 95% confidence intervals.

such novel H3K4me3 peaks were deposited across the genome of wild-type cells, translating to at least 1 extra peak every 10 CPDs or every 500 nucleosomes. The rather low ratio of H3K4me3 peaks to CPDs is consistent with the slow kinetics of CPD repair (see Fig. 1c). As the only exception, the H3K4me3 peak density did not change in the promoters

of active genes, where this histone mark is already enriched under constitutive conditions. The UV-induced formation of a surplus of H3K4me3 peaks at all other genomic features was largely dependent on ASH1L, as ASH1L-deficient cells had few such extra histone methylations (Fig. 3b, right row of bar plots). Representative H3K4me3-ChIP-seq profiles with a peak generated after UV irradiation in wild-type but not in ASH1L[−/−] cells are shown in Fig. 3c. On average, these de novo H3K4me3 marks, hereafter designated "novel ASH1L-deposited H3K4me3 peaks", extended over a width of 654 base pairs.

To confirm that the methyltransferase activity of ASH1L is responsible for H3K4me3 marks in response to UV light, the Flag-tagged CTD of ASH1L was expressed in ASH1L[−/−] cells to derive genome-wide CTD-ChIP-seq tracks using anti-Flag antibodies. We found that the recruitment of CTD upon UV exposure is significantly increased at all chromatin regions (Supplementary Fig. 5c). Moreover, a positional correlation between H3K4me3 peaks and CTD-ChIP-seq tracks, as outlined in Supplementary Fig. 5d, demonstrated that CTD is enriched at sites of novel ASH1L-deposited H3K4me3 peaks (Fig. 3d), thereby directly linking the methyltransferase enzyme occupancy to the deposition of methylation marks.

## ASH1L primes chromatin for XPC relocations

To determine how the ASH1L-deposited H3K4me3 marks influence XPC, the GG-NER initiator known to interact tightly with chromatin[48], ChIP-seq analyses were conducted in wild-type, XPC[−/−] and ASH1L[−/−] cells using an anti-XPC antibody (with XPC[−/−] cells included to determine background reads, Supplementary Fig. 5e). Reflecting the constitutive binding of XPC to undamaged DNA, XPC-ChIP-seq reads were detected throughout different genomic features even in the absence of UV damage (Supplementary Fig. 5f), supporting the view that chromatin is constitutively covered by XPC protein.

Despite the low nuclear mobility of XPC protein[48] and its prominence on chromatin, there were changes in XPC occupancy after UV irradiation at 20 J m[−2], which as outlined above is expected to generate -1 CPD per 10 kilobase pairs. In particular, some XPC was temporarily released from gene promoters and other genomic locations (gene bodies and intergenic regions) at 1 h after the UV challenge (Supplementary Fig. 5g), indicating that a fraction of XPC is diverted away from constitutive binding sites to allow for redistributions to damaged DNA. Preferred targets of this transiently mobilized fraction included the sites of novel ASH1L-deposited H3K4me3 peaks, as demonstrated by positional correlations between H3K4me3-ChIP-seq and XPC-ChIP-seq tracks (Fig. 3e). At 3 h after the UV pulse, the XPC occupancy of different genomic features tended to normalize to baseline levels, except for the sites of novel ASH1L-deposited H3K4me3 peaks in wild-type cells, where the XPC occupancy further increased between 1 and 3 h following irradiation (Supplementary Fig. 5g). From these findings, we conclude that some XPC moves from undamaged to damaged DNA and that the H3K4me3 addition by ASH1L-MRG15 primes nucleosomes for this damage-dependent XPC redistribution. In ASH1L[−/−] cells, such guidance by H3K4me3 is missing, resulting in XPC being misplaced to lesions that, as shown below, are not amenable to the downstream TFIIH complex.

## Enhanced excision at ASH1L-deposited H3K4me3 peaks

We next tested whether the novel H3K4me3 marks, which regulate XPC relocations following UV irradiation, translate to enhanced repair. For this purpose, induction and excision of UV lesions were monitored by high-sensitivity damage-sequencing (HS damage-seq), involving the enrichment of sequences containing CPDs and their detection using a high-fidelity DNA polymerase[6,49]. Cells were UV-irradiated as above to induce -1 CPD per 10 kilobase pairs. This UV exposure generated similar lesion frequencies (determined as CPDs per kilobase pairs) in wild-type and ASH1L[−/−] cells and across different genomic features (Supplementary Fig. 5h). CPD excision was monitored over time by the

progressive disappearance of sequencing reads. During the initial 3-h period, the global extent of CPD repair from sites of novel ASH1L-deposited H3K4me3 peaks was ~threefold higher in wild-type compared to ASH1L[−/−] cells and this repair deficiency of ASH1L[−/−] cells extended to each individual genomic feature (Fig. 3f). Data normalization to account for the varying proportion of DNA across the different features (see inset of Fig. 3f) showed that the reported[5,6] faster repair of both active and poised promoters compared to gene bodies and intergenic regions equally applies to H3K4me3 sites (Supplementary Fig. 5i).

The excision rates of Fig. 3f are influenced by an overlapping TC-NER pathway. However, intergenic regions undergo minimal transcription and, consequently, low TC-NER activity. This genomic feature was, therefore, further dissected to prove that GG-NER activity is most effective at novel ASH1L-deposited H3K4me3 peaks. For this purpose, we determined in wild-type cells the disappearance of CPD-related reads during the first 3 h of repair in 654-base pair long bins (equivalent to the average width of ASH1L-deposited H3K4me3 peaks) in three sub-features of euchromatic intergenic regions: (1) at sites of H3K4me3 peaks deposited by ASH1L following UV irradiation, (2) at sites of H3K4me3 peaks generated after UV irradiation independently of ASH1L, also found in ASH1L[−/−] cells, and (3) at random positions not containing H3K4me3 peaks. A comparison between these three genomic sub-features showed that CPD excision during the 3-h recovery was strongly confined to ASH1L-deposited H3K4me3 peaks; much less CPD excision took place elsewhere, i.e., outside of ASH1L-deposited peaks (Fig. 3g). This finding is further highlighted by relating the position of H3K4me3 peaks to CPD excision as outlined in Supplementary Fig. 5d. In euchromatic intergenic regions of wild-type cells, a clear positional correlation is observed between the centers of novel ASH1L-deposited H3K4me3 peaks and the locations of subsequent CPD excision (Fig. 3h, left panel). This correlation is missing with H3K4me3 peaks that arise genome-wide after UV irradiation but independently of ASH1L (Fig. 3h, middle panel) or with preexisting H3K4me3 peaks (Fig. 3h, right panel).

We next extended this analysis of H3K4me3 peaks and CPD excision to the full genome. First, there was at the level of the entire genome a correlation between the initial CPD formation upon UV exposure and the subsequent addition of the -55,000 H3K4me3 peaks by ASH1L (Fig. 4a). Second, CPD excision during the 3-h repair period after UV irradiation was strikingly elevated at the center of these ASH1L-deposited H3K4me3 peaks (Fig. 4b). Third, the preferential repair detected at H3K4me3 marks resulted in a pronounced dip in the distribution of the remaining CPDs (after 3 h of repair) within the sites of novel ASH1L-deposited H3K4me3 peaks (Fig. 4c). This positional correlation of H3K4me3 addition, CPD excision and remaining CPDs was not observed at random sequences from the remaining genome outside the sites of novel ASH1L-deposited H3K4me3 peaks (Fig. 4d–f).

Because ASH1L may also generate H3K36me2[28,32,44], we next conducted H3K36me2-ChIP-seq experiments. However, none of the chromatin features showed a substantial increase in the overall H3K36me2 peak density in response to UV irradiation (Supplementary Fig. 6a, b) comparable to the UV-induced increments described for H3K4me3 in Fig. 3b. Notably, novel H3K36me2 peaks detected after UV irradiation were added preferentially at sites located away from CPDs (Supplementary Fig. 6c, left panel) and the excision of CPDs did not correlate with such novel H3K36me2 peaks (Supplementary Fig. 6c, right panel). Moreover, considering that the CTD of ASH1L is recruited preferentially to CPD sites (Supplementary Fig. 6d, left panel) and that its position correlates with enhanced excision (Supplementary Fig. 6d, right panel), we conclude that the addition—at UV lesions—of H3K4me3 (but not H3K36me2) by ASH1L constitutes the critical mark for the repartitioning of XPC leading to GG-NER hotspots. In the absence of ASH1L or MRG15, this epigenetic mark is missing and, consequently, XPC is misplaced.

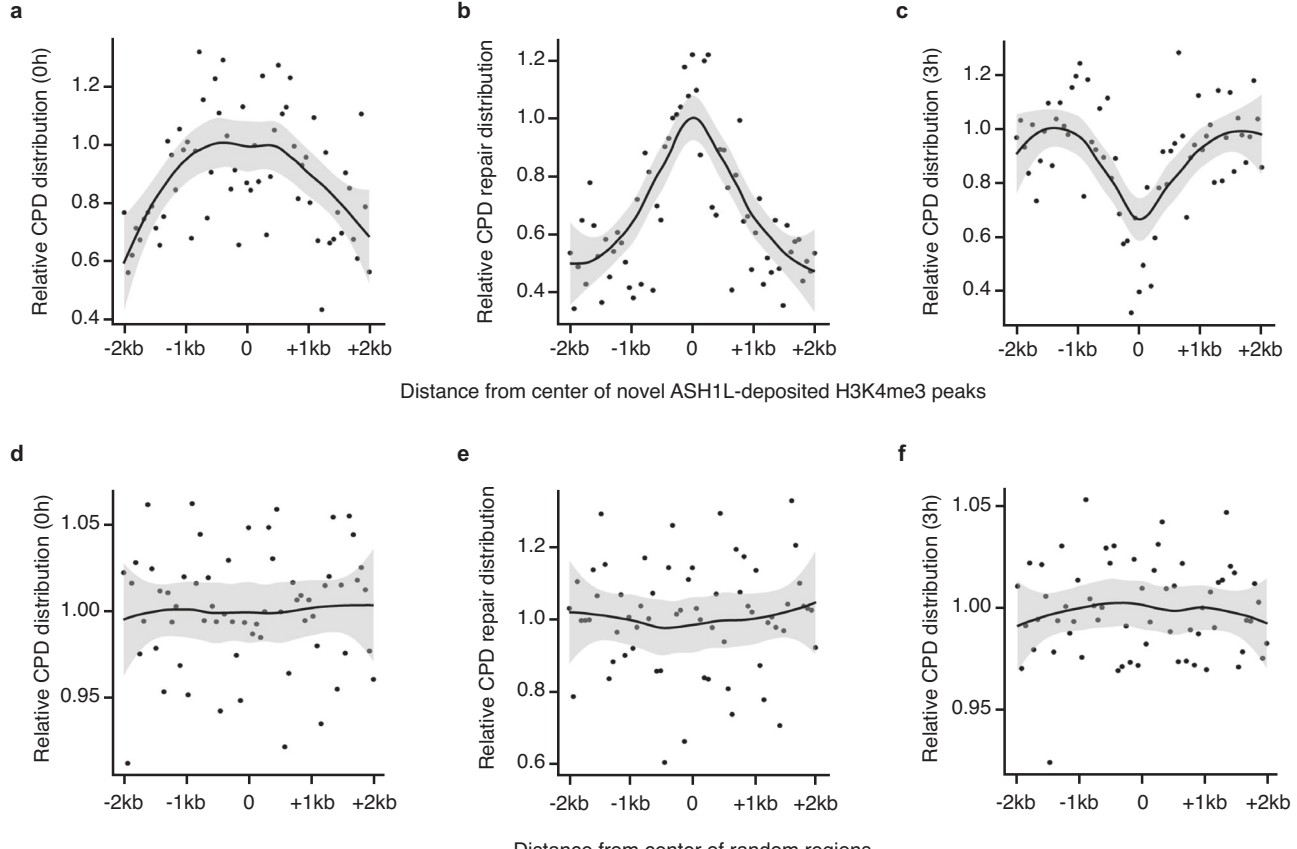

**Fig. 4 | CPD excision takes place mainly at de novo H3K4me3 marks.** Genome-wide positional correlation demonstrating that H3K4me3 is added preferentially at CPD sites to stimulate excision. Shaded areas around each CPD distribution line represent 95% confidence intervals. **a** In the first 3 h after UV irradiation, ASH1L deposits H3K4me3 peaks (determined by H3K4me3-ChIP-seq) mainly at sites of CPD induction (determined by HS damage-seq). The ~55,000 genomic sites harboring de novo ASH1L-dependent H3K4me3 peaks deposited post UV and their flanking sequences were subdivided into 64-bp bins as illustrated in Supplementary Fig. 5d. The mean initial abundance of CPDs is indicated for each of these bins. **b** During the

3-h repair, CPDs are excised preferentially from the center of H3K4me3 peaks deposited by ASH1L after UV irradiation. CPD excision is shown as the mean disappearance of CPDs in each of the 64-bp bins within novel ASH1L-deposited H3K4me3 peaks. **c** Hotspots of repair at H3K4me3 peaks translate to cold spots of remaining CPDs. Shown are the CPDs remaining after 3 h of repair in each 64-bp bin around ASH1L-deposited H3K4me3 peaks. The dip at the peak centers is indicative of preferential CPD repair. **d–f** Corresponding control plots for (**a–c**) using 64-bp bins encompassing random regions outside the de novo H3K4me3 peaks. In these control plots, zero values correspond to the center of the randomly selected regions.

## ASH1L promotes damage verification by TFIIH

To understand why the misplacement of XPC in ASH1L$^{-/-}$ cells dampens repair, UV lesion spots were induced in the nuclei of wild-type, ASH1L$^{-/-}$ and XPA$^{-/-}$ cells to monitor the recruitment of XPC to DNA damage 1 and 3 h after irradiation. This method allows for a close assessment of time- and damage-specific interactions. At 1 h after irradiation, there is no difference in XPC levels at UV spots in the different genetic backgrounds (Fig. 5a). In wild-type cells, XPC levels in UV spots decreased at 3 h relative to the 1-h time point, reflecting ongoing repair. However, XPC persisted abnormally in the UV spots of both ASH1L$^{-/-}$ and XPA$^{-/-}$ cells at 3 h after irradiation (Fig. 5a), with ASH1L$^{-/-}$ and XPA$^{-/-}$ cells displaying bright XPC spots co-localizing with CPDs (Fig. 5b). The retention of XPC in XPA$^{-/-}$ cells is caused by the absence of XPA protein, which is a downstream factor needed for NER to proceed after lesion recognition. The retention of XPC in ASH1L$^{-/-}$ cells thus indicates that ASH1L, in addition to guiding XPC relocations (Fig. 3e), is involved in a downstream reaction that facilitates the XPC turnover at lesion sites. Notably, this abnormal XPC retention depends on prior CPD detection by the DDB2 lesion receptor, as the appearance of XPC in UV lesion spots 3 h after irradiation is abolished by DDB2 depletion (Supplementary Fig. 7a, b).

Two lines of evidence indicate that the methyltransferase activity of ASH1L-MRG15 resulting in the de novo placement of H3K4me3 is indispensable to support the XPC turnover at lesion sites. First,

depletion of MRG15 by siRNA transfection of ASH1L-proficient cells leads to an abnormal XPC retention in UV lesion spots at the 3-h time point (Fig. 5c, see Fig. 5d for representative images), exactly as observed in ASH1L$^{-/-}$ cells. Second, rescue experiments showed that the retention of XPC at UV spots is reversed by transfection of ASH1L$^{-/-}$ cells with the vector coding for the active CTD fragment, but not with empty vector or the vector coding for CTD$_{inactive}$ (Fig. 5e, representative images in Fig. 5f).

XPC delivers DNA lesions to TFIIH, which in turn verifies the damage and induces the recruitment of XPA, RPA and the endonucleases making dual DNA incisions[11,17,50]. Interruption of this XPC – > TFIIH – > XPA –> endonuclease pathway in ASH1L$^{-/-}$ cells (due to the missing ASH1L protein) may lead to XPC persisting on DNA lesions without recruitment of the TFIIH complex. This hypothesis was tested by monitoring the nuclear distribution of XPD (the DNA helicase subunit of TFIIH) at 3 h after UV irradiation, i.e., at the time point when XPC was found to persist on lesions. Compared to wild-type controls, ASH1L$^{-/-}$ cells were characterized by a markedly reduced XPD recruitment to damage spots (Fig. 5g). Representative images illustrating the impaired engagement of XPD in ASH1L$^{-/-}$ cells are shown in Fig. 5h. In contrast, the XPD subunit accumulated in the UV spots of XPA$^{-/-}$ cells (Fig. 5g, h), consistent with XPA being involved downstream of TFIIH in the NER pathway. Again, two lines of evidence indicate that the methyltransferase activity of ASH1L-MRG15 and, hence, the placement

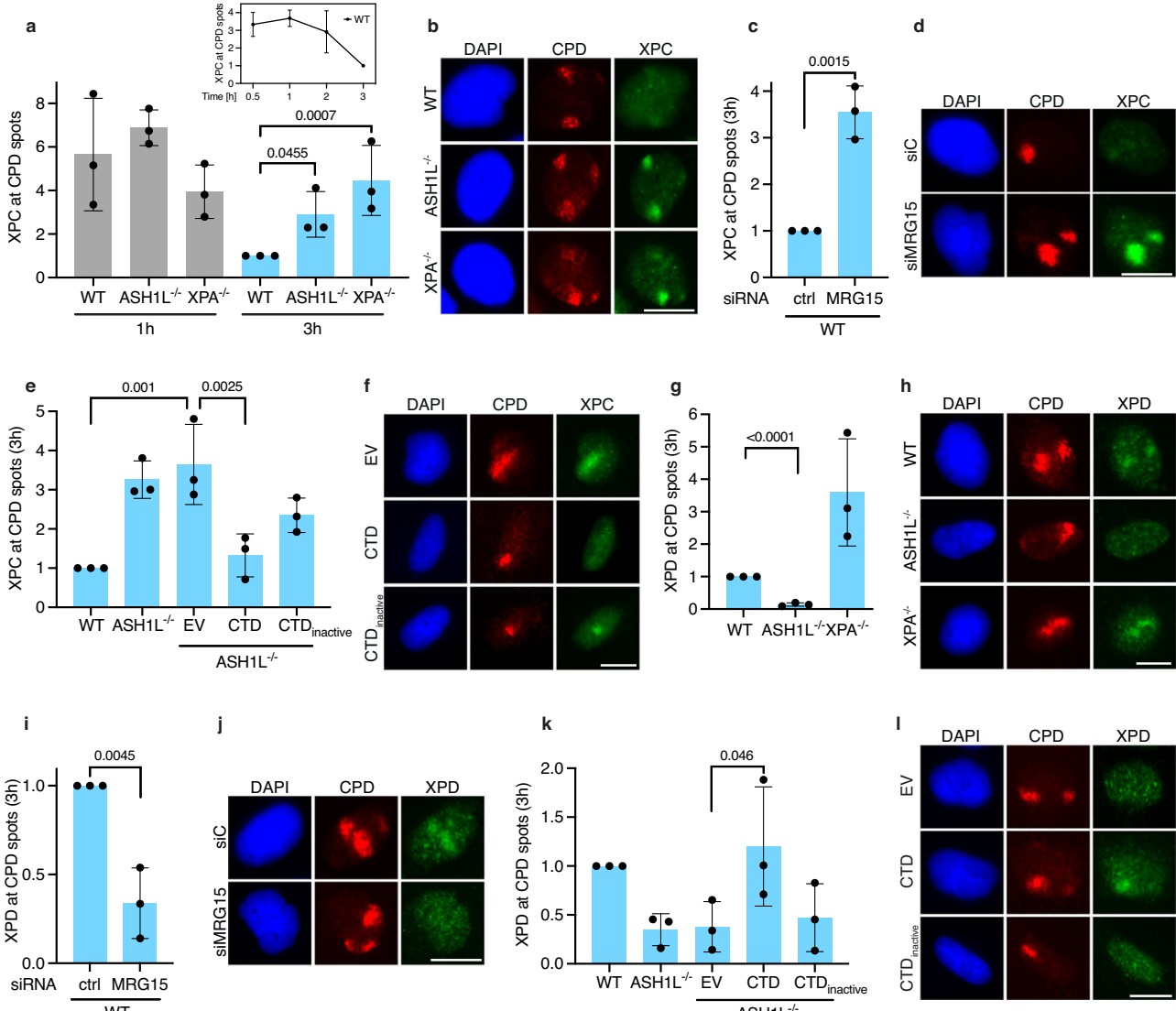

**Fig. 5 | ASH1L promotes TFIIH recruitment.** Without ASH1L-MRG15, XPC persists on lesions failing to recruit TFIIH. **a** XPC levels in UV spots of WT, ASH1L$^{-/-}$ and XPA$^{-/-}$ cells 1 and 3 h after UV irradiation. Quantification of three independent experiments normalized to the XPC level in WT cells 3 h post UV (inset: time course of XPC in UV spots); *P* values were determined by one-way ANOVA. **b** Fluorescence images showing that, 3 h post UV, XPC persists in damage spots of ASH1L$^{-/-}$ and XPA$^{-/-}$ cells relative to WT controls. **c** MRG15 depletion causes XPC persistence in damage spots 3 h post UV. Quantification of three independent experiments relative to control siRNA (two-tailed *t* test). **d** Images showing that, 3 h post UV, XPC persists in damage spots of MRG15-depleted cells. **e** The persistence of XPC in damage spots (3 h post UV) is reversed in ASH1L$^{-/-}$ cells by CTD, but not CTD$_{inactive}$ or empty vector (EV). Quantification of three independent experiments (one-way ANOVA). **f** Images showing that, 3 h post UV, the XPC

persistence in damage spots is reversed by CTD. **g** ASH1L$^{-/-}$ cells fail to recruit XPD to lesions 3 h post UV. Quantification of three independent experiments relative to XPD levels in WT cells (one-way ANOVA). **h** Images showing the defective XPD recruitment in ASH1L$^{-/-}$ cells 3 h post UV. **i** MRG15 depletion impairs the XPD recruitment to lesions 3 h post UV. Quantification of three independent experiments relative to control siRNA (two-tailed *t* test). **j** Images showing that, 3 h post UV, MRG15-depleted cells are impaired in the XPD relocation to lesions. **k** The recruitment of XPD to damage spots (3 h post UV) is restored in ASH1L$^{-/-}$ cells by CTD, but not CTD$_{inactive}$ or EV. Quantification of three independent experiments (one-way ANOVA). **l** Images showing that, 3 h post UV, the XPD recruitment to damage spots is rescued by CTD. **a**, **c**, **e**, **g**, **i**, **k** represent mean values ± SD. Scale bars in (**b**, **d**, **f**, **h**, **j**, **l**) correspond to 15 μm.

of H3K4me3 marks guiding XPC relocations is indispensable for the subsequent engagement of the XPD helicase. First, depletion of MRG15 by transfection with siRNA dampened the redistribution of XPD to UV lesions (Fig. 5i, representative images in Fig. 5j), as observed in ASH1L$^{-/-}$ cells. Second, the XPD recruitment to UV lesions was restored in rescue experiments when ASH1L$^{-/-}$ cells were transfected with the vector coding for active CTD but not with empty vector or the vector coding for CTD$_{inactive}$ (Fig. 5k; representative images in Fig. 5l). A markedly reduced recruitment to damage spots in ASH1L$^{-/-}$ compared to wild-type cells was also observed for XPB (another TFIIH subunit) and XPA (a further downstream NER factor; Supplementary Fig. 7c–f).

## The FACT chaperone mediates TFIIH recruitment

It was intriguing to find in Fig. 3g that H3K4me3 peaks that arise independently of ASH1L (genomic feature No. 2) are unable to stimulate NER activity to the same extent as do ASH1L-deposited H3K4me3 peaks (feature No. 1). A possible explanation for this difference is provided in Fig. 5, showing that, besides depositing H3K4me3 to guide XPC relocations, the ASH1L-MRG15 methyltransferase also facilitates the follow-up handover from XPC to TFIIH. Given that XPC is known to interact directly with TFIIH subunits[51–53], why would ASH1L-MRG15 be needed for the TFIIH recruitment?

One hypothesis was that the ASH1L-MRG15 complex induces chromatin relaxation that favors DNA accessibility. The assay for transposase-accessible chromatin using sequencing (ATAC-seq, Supplementary Fig. 8a, b) indeed demonstrated an increased substrate accessibility after UV irradiation compared to unchallenged cells, particularly in gene bodies and intergenic regions (Supplementary Fig. 8c) and, to a minor extent, at sites of novel ASH1L-deposited H3K4me3 peaks (Supplementary Fig. 8d). However, this improved access to DNA was indistinguishable between wild-type and ASH1L$^{-/-}$ cells, suggesting that the ASH1L-MRG15 complex promotes the XPC-to-TFIIH transition by another mechanism. Therefore, immunoprecipitations (IPs) were conducted to screen for additional MRG15- and ASH1L-associating factors involved in the XPC-to-TFIIH exchange.

U2OS cells were transfected with constructs coding for MRG15 or CTD conjugated with Flag. Solubilized lysates of cells expressing Flag-MRG15 or Flag-CTD were subjected to IPs using anti-Flag antibodies. By mass-spectrometric analyses of the eluates, we searched for factors that interact with both MRG15 and CTD. Among the common interactors linked to chromatin dynamics (Fig. 6a), SPT16 (SuPpressor of Ty 16) was an intriguing candidate in view of its nucleosome-organizing function. This same protein was also detected in complexes pulled down from solubilized U2OS cell lysates using an anti-endogenous XPC antibody (the list of interactors is provided in Supplementary Table 1). Together with SSRP1 (Structure-Specific Recognition Protein 1), SPT16 forms a histone chaperone known as FACT that, by nucleosome assembly and disassembly, promotes transcription[54], transcription restart after DNA damage[55], DNA damage signaling[56] and TC-NER[57]. Because SPT16 and SSRP1 have never before been reported to associate with MRG15 or ASH1L, solubilized lysates of cells expressing Flag-MRG15 or Flag-CTD were again subjected to IP using anti-Flag antibodies, but subsequently analyzed by immunoblotting. We found that SPT16 co-immunoprecipitates with both MRG15 and CTD (Supplementary Fig. 9), thus confirming that FACT co-exists in soluble protein complexes with ASH1L-MRG15.

Next, the two subunits of FACT were depleted (separately or in combination) in U2OS cells by siRNA treatments. UDS measured in UV lesion spots as outlined in Fig. 1d revealed that both SPT16 and SSRP1 are implicated in the GG-NER reaction (Fig. 6b, see Fig. 6c for representative images). Knowing that FACT stimulates GG-NER activity, we next tested whether SPT16, as one of the two FACT subunits, is essential for the XPC-to-TFIIH exchange at DNA lesions. To this end, the nuclear distributions of XPC (Fig. 6d, e) and XPD (Fig. 6e, f) were compared in control and SPT16-depleted cells 3 h after the induction of UV spots. Upon SPT16 depletion, XPC persisted at lesion sites and XPD recruitment was reduced, establishing that an SPT16 deficiency abrogates the XPC-to-TFIIH handover exactly as observed in the absence of ASH1L or MRG15.

SPT16 relocated to the UV spots of wild-type cells 3 h but not 1 h after UV radiation (Fig. 6g, see Fig. 6h for representative images). This latency suggests that SPT16 is needed to process recalcitrant lesions that are not readily removed, i.e., CPDs within nucleosomes. In any case, the recruitment of SPT16 observed in wild-type cells 3 h after UV irradiation was missing in the ASH1L$^{-/-}$ background (Fig. 6g, h). An involvement of MRG15 and ASH1L in the SPT16 recruitment to UV lesions was confirmed in wild-type cells after depleting either factor by siRNA treatments (Fig. 6i). This SPT16 recruitment to UV lesions was also diminished by DDB2 depletion, providing further support for a role of the FACT chaperone in the GG-NER reaction. Moreover, in ASH1L$^{-/-}$ cells the recruitment of SPT16 to UV lesion spots was rescued by transfection with the CTD vector (Fig. 6j; see representative images in Fig. 6k), consistent with the conclusion that ASH1L-MRG15 undergoes interactions with SPT16 during the GG-NER reaction. In these rescue experiments, both CTD and CTD$_{inactive}$ restored the SPT16 recruitment to UV lesions, implying that the histone methyltransferase activity of ASH1L-MRG15 is needed for the repartitioning of XPC in

response to DNA damage, but not for the successive engagement of FACT.

## Discussion

Contrary to the notion that nucleosomes—the basic structural repeats of chromatin—pose a barrier to DNA damage recognition, XPC and DDB2, the two factors launching GG-NER activity, can detect damaged sites even when the DNA is wrapped around histones in nucleosomes. XPC, which is the initiator of GG-NER activity, resides constitutively in chromatin[46,48], where it associates with histones to expediate DNA damage recognition[27,58,59]. DDB2, whose role is to recruit the XPC subunit to UV photoproducts, similarly detects damaged substrates within nucleosomes[60,61]. The case of CPDs is important because these abundant lesions arise evenly along nucleosome arrays, with the consequence that most CPDs of a UV-damaged genome are embedded in nucleosome cores and only a minor fraction of these lesions appears in histone-free regions such as linker DNA joining nucleosomes[6,7,62]. Here, we describe the histone methyltransferase complex ASH1L-MRG15 as an essential GG-NER factor that ensures a rapid excision of CPDs by promoting the XPC-to-TFIIH transition in the context of nucleosome-embedded DNA damage (Figs. 1 and 5).

Several findings explain how the ASH1L-MRG15 complex stimulates DNA damage excision. First, rescue experiments in ASH1L$^{-/-}$ cells demonstrate that the intact methyltransferase activity of ASH1L is required to stimulate the GG-NER reaction. A loss-of-function mutation in the catalytic subdomain of ASH1L abolishes this repair function. A role of the methyltransferase activity of ASH1L in GG-NER is further demonstrated by the contribution of MRG15, which by interaction with ASH1L releases a self-inhibitory loop covering the catalytic site[42,63]. Upon depletion of MRG15, the capacity of ASH1L to support GG-NER activity is nearly completely abolished (Fig. 2).

Second, the genome-wide cartography of DNA lesions, occupancy of the CTD of ASH1L, histone methylation, as well as the XPC distribution, show that ASH1L-dependent H3K4me3 depositions, occurring preferentially at CPD sites, guide XPC relocations in response to UV damage. Through the placement of H3K4me3 and the recruitment of XPC to these de novo H3K4me3 marks, ASH1L diverts a fraction of the cellular XPC pool away from constitutive binding sites towards damaged sites permissive to the recruitment of TFIIH. Consequently, after UV irradiation, CPD excision occurs preferentially at the locations of these novel ASH1L-deposited H3K4me3 marks (Figs. 3 and 4). In the absence of ASH1L or MRG15, the GG-NER initiator XPC is misplaced within the genome such that it persists on UV lesions without being able to recruit the follow-up TFIIH complex (Fig. 5).

Third, ASH1L-MRG15 exploits the histone chaperone FACT to facilitate the lesion handover from the XPC sensor to the TFIIH verifier (Fig. 6). H3K4me3 marks or other chromatin modifications, although attracting XPC to damaged nucleosomes, are evidently not sufficient to reshuffle chromatin for the recruitment of follow-up GG-NER factors. A possible player in this process was the chromatin remodeler CHD1 (Chromodomain Helicase DNA-binding 1), which is recruited through H3K4 methylation during transcription[64,65]. Another candidate was the BAZ1A (Bromodomain Adjacent to Zinc finger domain 1A) chromatin remodeler that is recruited after H3K4 methylation by MLL1[45]. Unlike ASH1L, however, we found that CHD1[66] and MLL1 (Fig. 2) contribute little to CPD repair, prompting a screen for additional factors interacting with ASH1L, MRG15 and XPC. This approach disclosed FACT as a common interactor of the above three proteins and, indeed, a depletion of FACT subunits severely compromises the TFIIH recruitment and GG-NER activity (Fig. 6). An integrative model of how ASH1L uses the deposition of H3K4me3 and FACT to coordinate the engagement of XPC and TFIIH at UV lesions is depicted in Fig. 7.

In summary, the XPC-to-TFIIH handover leading to DNA damage excision depends on H3K4me3 and FACT, which are typical attributes

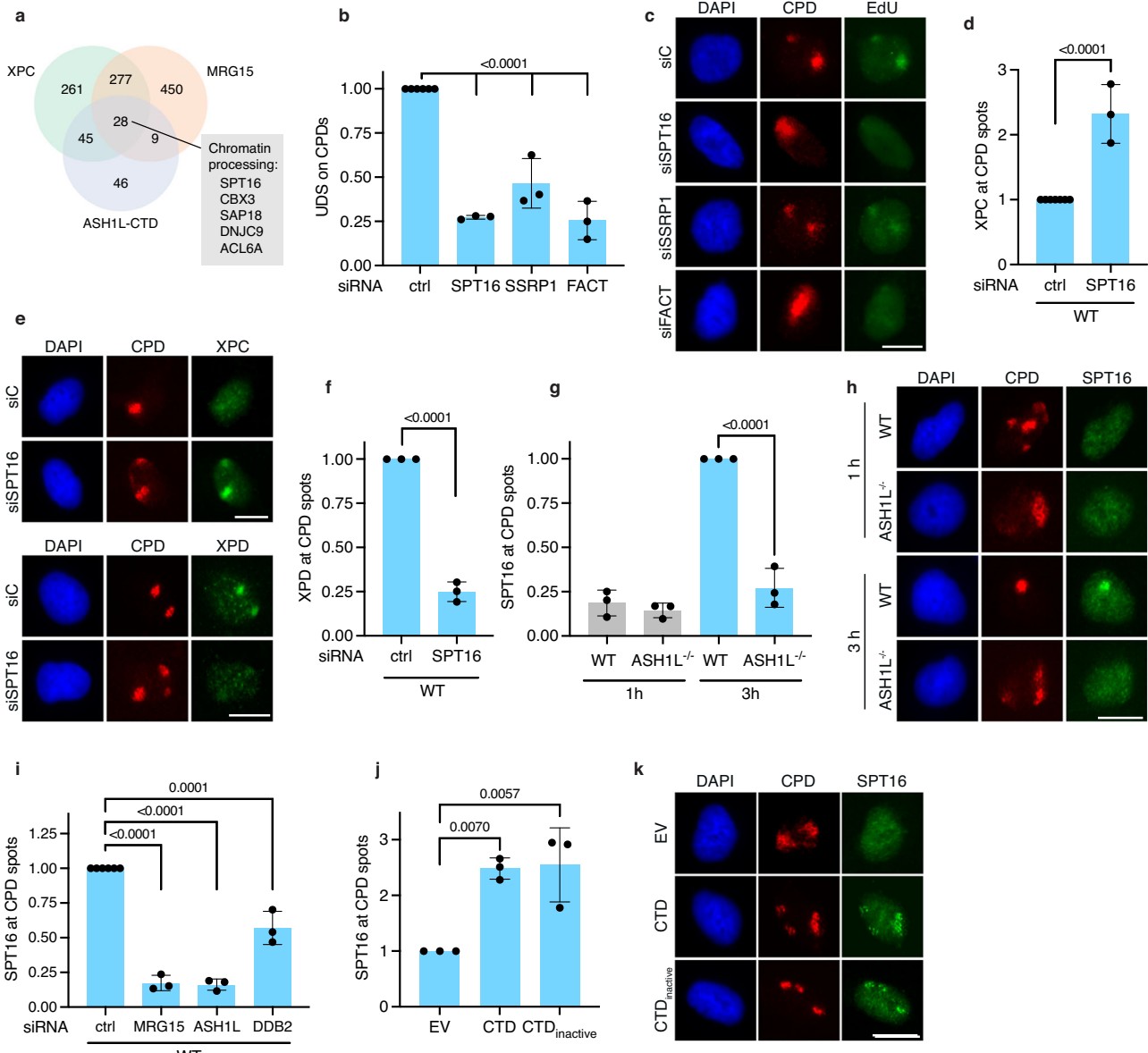

**Fig. 6 | MRG15-ASH1L recruits FACT for the XPC-to-TFIIH handover.** To deliver DNA lesions to TFIIH, ASH1L recruits the histone chaperone FACT. **a** Mass spectrometry analysis of immunoprecipitates revealing factors that interact with MRG15, CTD and XPC. Among five common interactors involved in chromatin processing, SPT16 was selected for further analyses. **b** Quantification of three independent UDS assays with WT cells subjected to the indicated siRNA treatments; ctrl, U2OS cells transfected with control siRNA; FACT, cells transfected with siSPT16 and siSSRP1 simultaneously. Mean values ± SD; $P$ values were determined by one-way ANOVA. **c** Immunofluorescence images illustrating the impaired UDS after depletion of FACT subunits, either individually (siSPT16 or siSSRP1) or in combination (siFACT); siC, cells transfected with control siRNA. **d** Persistence of XPC at UV lesion spots at 3 h post UV, following SPT16 depletion. Mean values ± SD ($n = 3$ independent experiments); two-tailed $t$ test. **e** Fluorescence images illustrating the persistence of XPC and the reduced recruitment of XPD at 3 h post UV, following SPT16 depletion. **f** Quantification of three independent experiments showing the impaired recruitment of XPD to UV lesion spots 3 h post UV, following SPT16 depletion. Mean values ± SD; two-tailed $t$ test. **g** Quantification of SPT16 levels in the UV lesion spots of WT and ASH1L$^{-/-}$ cells. Mean values ± SD ($n = 3$ independent experiments); one-way ANOVA. **h** Fluorescence images demonstrating the ASH1L-dependent recruitment of SPT16 to UV spots 3 h after irradiation. **i** The recruitment of SPT16 to UV lesion spots (3 h after irradiation) is diminished by siRNA-mediated depletions of MRG15, ASH1L or DDB2. Mean values ± SD ($n = 3$ independent experiments); one-way ANOVA. **j** Restoration of SPT16 recruitment to UV spots of ASH1L$^{-/-}$ cells, 3 h post UV, by transfection with vectors coding for CTD and CTD$_{inactive}$. EV, cells transfected with empty vector. Mean values ± SD ($n = 3$ independent experiments); one-way ANOVA. **k** Fluorescence images demonstrating the rescue of SPT16 recruitment to UV lesion by CTD and CTD$_{inactive}$. Scale bars in (**c, e, h, k**) correspond to 15 μm.

of transcriptional start sites. This finding thus indicates that ASH1L-MRG15 confers an active promoter-like histone code and nucleosome arrangement of chromatin to stimulate NER activity. Interestingly, the analogous NER process in bacteria relies heavily on the transcription-coupled subpathway, which accounts for most of the excision activity removing UV damage and other bulky lesions in *Escherichia coli*[67]. In human cells, instead, XPC initiates the distinctive GG-NER subpathway

that is independent of ongoing transcription[3,4]. With TFIIH, however, even this specialized GG-NER reaction takes advantage of a generic factor that is shared with transcription initiation. Our report identifies further elements (ASH1L, MRG15, H3K4me3 and FACT) shared with transcription, which are exploited not only in promoters but across the entire genome to convert damaged chromatin sites to a verifiable substrate compatible with TFIIH recruitment.

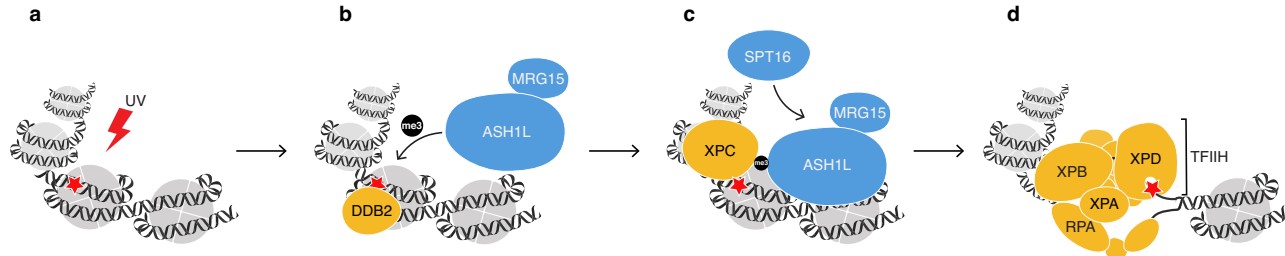

**Fig. 7 | Model of how ASH1L-MRG15 facilitates DNA damage verification.** This histone methyltransferase exploits a bimodal mechanism (through the sequential H3K4me3 and FACT deposition) to make DNA damage verifiable in nucleosomes. **a** UV irradiation generates CPDs across the genome including in histone-associated DNA. **b** ASH1L-MRG15 is recruited by the DDB1-cullin 4A-DDB2 complex (only the DDB2 lesion receptor is shown). By depositing H3K4me3, ASH1L primes nucleosomes for XPC binding, thus navigating this GG-NER initiator away from constitutive locations on native DNA towards damaged sites. In ASH1L$^{-/-}$ cells, this priming for repair is rescued by the enzymatically active CTD of ASH1L but not by CTD$_{inactive}$, indicating that the methyltransferase activity is indispensable to initiate the GG-NER reaction. This enzymatic function of ASH1L-MRG15, culminating in the deposition of H3K4me3, thereby acts as a caliper to gauge whether chromatin is amenable to large downstream NER factors, primarily the TFIIH complex. **c** As we previously reported[27], XPC protein interacts, via a β-turn motif embedded in its DNA-binding domain, with ASH1L-deposited H3K4me3. Upon recognition of H3K4me3-marked lesions by XPC, H3K4m3 serves as an anchor for ASH1L itself to remain associated (through its PHD finger[77]) with methylated nucleosomes, thus forming docking sites for the histone chaperone FACT (only the SPT16 subunit is shown). In ASH1L$^{-/-}$ cells, FACT recruitment is rescued by both active and inactive CTD, indicating that, at this step, a non-enzymatic matchmaker function of ASH1L-MRG15 is sufficient to couple FACT to the GG-NER machinery. **d** In turn, FACT supports the XPC-dependent recruitment of TFIIH through nucleosome rearrangements and, once loaded onto DNA, TFIIH deploys its XPD helicase subunit for damage verification by scanning the damaged strand[17–19]. This model accounts for the observation that, in the absence of the ASH1L enzymatic activity, XPC still detects DNA damage but fails to be escorted to chromatin sites that are amenable to the downstream TFIIH engagement. Because of this misplacement, XPC persists on lesion sites and, even in the presence of FACT, is unable to deliver the DNA substrate to the GG-NER machinery.

## Methods

### Cell lines and gene deletions
U2OS and HEK293T cells, obtained from American Type Culture Collection, were maintained in low glucose Dulbecco's Modifies Eagle Medium (Gibco, 21885-025) supplemented with 10% (v/v) fetal calf serum (FCS, Gibco, 10500-064). These cell lines were authenticated by short tandem repeat profiling and tested negative for mycoplasma.

The CRISPR/Cas9 system[68] was employed to disrupt the genes coding for ASH1L, XPA, XPC and XPF. Briefly, cells were transfected in 12-well plates with pSPCas9 plasmids harboring sgRNA sequences (Supplementary Fig. 1) using the FuGENE reagent (Promega). The puromycin selection (1 µg/ml, Gibco) started 24 h after transfection and lasted for several days. Individual colonies were isolated and deletions validated by PCR, sequencing and immunoblot analyses.

### Clonogenic survival
U2OS cells were seeded in 24-well plates (300 cells/well) and incubated overnight to allow for attachment. After UV-C irradiation, fresh medium was added, and the cells were left to recover for 7 days with intermittent medium changes. Thereafter, cells were washed with phosphate-buffered saline (PBS) and fixed in ice-cold methanol for 10 min. For blocking, 0.5× Intersept PBS Blocking Buffer (Licor, 927-70001) was distributed over each well followed by a 30-min incubation at room temperature. The cells were stained with 300 µl/well CellTag 700 Stain (Licor, 926-41090) diluted 1:500 in 0.5× Intersept PBS Blocking Buffer. After incubation (1 h) at room temperature on a horizontal shaker protected from light, the wells were washed twice for 5 min with PBS containing 0.05% (v/v) Tween 20 (PBS-T) and dried overnight at room temperature. Finally, the plates were scanned using the Odyssey CLx device (Licor) and fluorescence intensity was quantified with Image Studio v.5.2.5.

### Cell cycle analysis
U2OS cells were synchronized in 5 µM L-mimosine (Sigma, M0253) for 24 h. After UV-C irradiation, the cells were left to recover for 6 or 18 h with the addition of 10 µM EdU (Invitrogen, C10337) for the final 30-min incubation. The cells were collected and fixed in 4% (w/v) paraformaldehyde (pH 8.0) for 15 min at room temperature. After permeabilization in 1× saponin buffer (Invitrogen, C10337) for 15 min at room temperature, the Click-iT reaction was performed according to the manufacturer's protocol (Invitrogen). Thereafter, samples were incubated with 0.1 mg/mL RNase A (Thermo Scientific) for 30 min at 37 °C and the DNA was stained with 25 µg/mL propidium iodide (PI, Invitrogen) for 5 min. Using the BD LSR II Fortessa flow cytometer (10,000 cells per sample), PI signals were plotted against Alexa-488. Cell cycle distributions were evaluated using the FlowJo v.10.9 software.

### UV irradiation and induction of UV spots
Cell culture medium was removed, and cells were washed with PBS prior to UV-C irradiation at 254 nm using a germicidal lamp. For the induction of UV spots, polycarbonate filter membranes with 5-µm pores (Whatman, WHA10417406) were placed on the cells prior to irradiation at 100 J m$^{-2}$. Immediately after irradiation, fresh prewarmed culture medium was added and the cells were left to recover for the indicated times.

### CPD excision assay
To carry out the enzyme-linked immunosorbent assay (ELISA), U2OS cells were UV-C- irradiated at 10 J m$^{-2}$ and left to recover for different periods. Thereafter, cells were collected, and DNA extracted using the QIAamp DNA Blood Kit (QIAGEN). The DNA concentration was adjusted to 1 ng/µl and, for denaturation, samples were incubated at 100 °C for 10 min followed by immediate cooling on an ice-water bath for 15 min. The 96-well microtiter plates (Greiner) were coated with 0.003% (w/v) protamine sulfate (Sigma), dried overnight at 37 °C and loaded with 50 µl of DNA solution per well. Once coated with DNA, the plates were washed five times with PBS-T, blocked with 2% (v/v) FCS in PBS at 37 °C for 1 h, then washed again with PBS-T. Next, the plates were incubated with antibodies against CPDs (TDM-2; dilution of 1:1000 in PBS) for 30 min at 37 °C. These primary antibodies were detected by biotin-labeled F(ab')2 fragments from anti-mouse IgG (dilution 1:2000; see Supplementary Table 2 for the list of antibodies) added for 30 min at 37 °C. After washing the plates with PBS-T, 100 µl of a peroxidase streptavidin conjugate (dilution 1:10,000, Invitrogen, 434323) were added to each well. After 30 min at 37 °C, the plates were washed and the color reaction was started by the addition of 100 µl of substrate solution containing 0.5 mg ml$^{-1}$ o-phenylenediamine,

0.007% (v/v) H$_2$O$_2$ in citrate-phosphate buffer (50 mM Na$_2$HPO$_4$, 24 mM citric acid, pH 5.0). After stopping the reaction with 50 µl of 2 M H$_2$SO$_4$, absorbance was detected at 492 nm in a PLUS384 microplate spectrophotometer.

## Unscheduled DNA synthesis

Cells were grown on 13-mm glass coverslips to 80% confluency and UV-C-irradiated through micropore filters. To measure DNA repair synthesis in the UV spots[69], after a recovery of 3 h in standard medium, 10 µM EdU was added for 1 h. Next, cells were washed with PBS, pre-extracted in pre-extraction buffer [25 mM HEPES, pH 7.5, 50 mM NaCl, 1 mM EDTA, 3 mM MgCl$_2$, 300 mM sucrose, 0.5% (v/v) Triton X-100] for 2.5 min on ice, fixed in 4% (w/v) paraformaldehyde (pH 8.0) for 15 min at room temperature and washed in PBS-T twice for 10 min. DNA was denatured for 8 min in 0.07 M NaOH before incorporated EdU was coupled to Alexa Fluor 488 using the Click-iT reaction according to the manufacturer's instructions. After blocking in PBS with 20% (v/v) FCS for 30 min at 37 °C, the antibody against CPDs (TDM-2; dilution of 1:1000) was added for 1 h at 37 °C. Cells were washed for 20 min with PBS-T. DNA was stained with DAPI (0.2 µg/ml) and secondary antibody (Supplementary Table 2). After washing with PBS-T for another 20 min, the coverslips were mounted on glass slides with ProLong Gold Anti-fade Mountant (Invitrogen). Images were taken with a fluorescence inverted microscope (DMI6000 B at 63x magnification with oil immersion lens) and analyzed with ImageJ v.2.8.0 software. EdU incorporation was quantified in at least 100 cells by determining fluorescence intensity in the CPD spots after subtraction of background nuclear intensity. S-phase cells displaying high nuclear EdU fluorescence were excluded.

## RNA interference and quantitative PCR

Cells were transfected with 16 nM siRNA in complete medium (see Supplementary Table 3 for the list of siRNA sequences) using the Lipofectamine RNAiMAX transfection reagent (Invitrogen). To control for cross-targeting, either two individual siRNAs or siRNA pools pre-designed by the SMARTselection algorithm (version pSeven 6.16) were used. Experiments were carried out 48 h after siRNA transfections. For RT-PCR determinations, RNA was extracted from cells using the RNeasy Mini Kit (Qiagen). Concentration and quality were assessed in a Nanodrop 2000c (Thermo Scienfic) spectrophotometer. With 1 µg RNA, reverse transcription was performed using the iScript cDNA Synthesis Kit (Bio-Rad). For quantitative RT-PCR, the reactions were carried out in duplicates using the KAPA SYBR FAST system (Roche) and amplified using the CFX384 Real-time C1000 Touch Thermal Cycler (see Supplementary Table 4 for the list of PCR primers). Relative RNA levels were calculated as $2^{-\Delta\Delta CT}$, normalized to the housekeeping internal standards glycerinaldehyde-3-phosphate-dehydrogenase or b2-microglobulin in comparison to control conditions.

## Plasmid cloning and transfection

Cloning was performed using the GATEWAY system from Invitrogen. The GATEWAY vector pDEST-CMV-N-Flag was engineered to express fusion proteins with an amino-terminal nuclear localization signal (from c-myc) and a carboxy-terminal 2xHA tag. The complementary DNA encoding the CTD of ASH1L was PCR-amplified (see Supplementary Table 4 for primers) from a human ASH1L ORF clone (Biocat, 55870). The loss-of-function mutant (CTD$_{inactive}$) was generated by mutagenesis following the QuickChange II site-directed mutagenesis manual. To obtain tagged MRG15, a recombination reaction was performed between the pDEST-CMV-N-Flag and a human MRG15 ORF clone (Biocat, 10933). Plasmids were propagated in CopyCutter EPI400 competent *E. coli* (Epicentre) and isolated using the Qiagen Plasmid Maxi Kit. For plasmid transfection, U2OS cells were grown to 80% confluency and incubated with the JetOPTIMUS reagent (Poly-plus) for 24 h.

## Cell lysis

Whole-cell lysis was carried out using NETN lysis buffer [100 mM NaCl, 20 mM Tris-HCl (pH 8.0), 0.5 mM EDTA, 0.5% (v/v) NP-40, supplemented freshly with EDTA-free protease inhibitor cocktail (Roche), 0.1% (w/v) benzonase and 1 mM MgCl$_2$] on a rotating wheel for 30 min at 4 °C. Samples were sonicated for 3 cycles (30 s on/30 s off) using the Bioruptor Plus sonication device (Diagenode) and centrifuged at 16,000 × g for 20 min at 4 °C. Supernatants were collected and protein concentrations determined using the Bradford reagent (Sigma).

## Immunoprecipitations

After two washing steps on ice with TBS (50 mM Tris-HCl, pH 7.4, 150 mM NaCl, and two washing steps with NETN, 40 µl anti-Flag M2 affinity gel (Sigma) were incubated with 500 µg of whole-cell lysate overnight at 4 °C on a turning wheel. Beads were washed twice for 10 min with TBS-0.05% (vol/vol) Tween 20 and twice for 10 min with TBS followed by centrifugation (3 min, 300×g). Protein elution was carried out by the incubation of the beads with 3x Flag peptide (Sigma) as the competitor for 40 min at 4 °C, followed by centrifugation at 5000×g for 30 s. Supernatants were analyzed by mass spectrometry (see Supplementary Methods) or transferred to loading buffer [60 mM Tris-HCl, pH 6.8, 10% (v/v) glycerol, 2% (w/v) sodium dodecyl sulfate (SDS), 1.25% (v/v) β-mercaptoethanol, 0.01% (w/v) bromophenol blue] for electrophoretic separation on polyacrylamide gels.

## Immunoblotting

Proteins were separated on 4–20% Criterion TGX stain-free precast gels (Bio-Rad, 5678093) at 150 V for 60 min and transferred to a 0.2-µm nitrocellulose membrane (Bio-Rad, 1704159) using the Trans-Blot Turbo transfer device (Bio-Rad, 7 min at constant 5 A). Membranes were blocked, incubated in primary antibodies overnight, washed, incubated with HRP-conjugated secondary antibodies for 1 h, washed and developed using the ChemiDoc Imaging System (Bio-Rad). See Supplementary Table 2 for antibodies and dilutions.

## In situ protein immunofluorescence

Cells were grown on 13-mm glass coverslips to 80% confluency and UV-C-irradiated through micropore filters as described above. After pre-extraction, cell fixation and washing in PBS-T, DNA was denatured in 0.07 M NaOH for 8 min. For blocking, cells were incubated in 20% (v/v) FCS for 30 min at 37 °C. Primary antibodies (Supplementary Table 2), diluted in 5% (v/v) FCS, were added to the coverslips and incubated for 1 h at 37 °C. The coverslips were washed in PBS-T twice for 10 min before incubation with secondary antibodies and DAPI for another 1 h at 37 °C. After washing with PBS-T, the coverslips were mounted on glass slides with ProLong Gold Antifade Mountant (Invitrogen). Fluorescence microscopy was performed using the Leica DMI6000 B at ×40 magnification.

## ChIP-seq assays

Triplicate libraries were generated for H3K4me3-, H3K36me2-, CTD-, and XPC-ChIP-seq according to a modified protocol[70]. U2OS cells (wild-type, ASH1L$^{-/-}$, ASH1L$^{-/-}$ transfected with CTD, and, as background controls, XPC$^{-/-}$ and ASH1L$^{-/-}$ transfected with empty vector) were grown in 10-cm dishes to 80–90% confluency, UV-C-irradiated at 20 J m$^{-2}$, left to recover for 1 h or 3 h, crosslinked in 1% (v/v) formaldehyde for 10 min, and quenched with 0.125 M glycine followed by two PBS washes. Cells collected with a cell scraper were incubated in cell lysis buffer [5 mM PIPES, 85 mM KCl, 1% (v/v) NP-40 and 1× EDTA-free protease inhibitor cocktail (Roche, 11836153001)] for 15 min on ice, then centrifuged at 400 g for 5 min at 4 °C. Cell pellets were transferred to sonication tubes and incubated in nuclear lysis buffer [50 mM Tris-HCl, pH 8.0, 10 mM EDTA, 1% (w/v) SDS and 1× protease inhibitor cocktail] for 30 min on ice, followed by flash freezing, thawing, sonication (Bioruptor Plus) in an ice-water bath for 20 cycles (30 s

on/30 s off), centrifugation at 10,000×*g* for 10 min at 4 °C, snap freezing and storage at −80 °C. To quantify the chromatin, 10-µl aliquots were diluted tenfold with ChIP elution buffer [50 mM NaHCO₃, 1% (w/v) SDS)], supplemented with 1 µl proteinase K and 12.5 µl proteinase K mix [2 M NaCl, 0.4 M Tris-HCl (pH 8.0), 0.1 M EDTA], and incubated at 50 °C for 3 h and at 65 °C overnight. Following the addition of 10 U of RNase (ThermoFisher Scientific, EN0601), the quantification samples were incubated at 37 °C for 20 min. DNA was purified with the QIAquick PCR Purification Kit (Qiagen, 28106), quantified by NanoDrop and visualized by agarose gel electrophoresis. From the crosslinked and sonicated chromatin, 30 µg (for H3K4me3, H3K36me2 and CTD) or 200 µg (for XPC) were re-thawed, diluted fivefold in IP dilution buffer [50 mM Tris-HCl, pH 7.4, 150 mM NaCl, 1% (v/v) NP-40, 0.25% (w/v) deoxycholic acid, 1 mM EDTA and 1× protease inhibitor cocktail] and incubated overnight at 4 °C on a rotating wheel with the corresponding antibody (9 µg of anti-H3K4me3-ChIP-grade, Abcam, ab8580; 4 µg of anti-H3K36me2-ChIP-grade, Abcam, ab9049; 5 µg of anti-FLAG M2, Sigma, F1804; 20 µg of anti-XPC, Invitrogen, PA5-97019). The next day, samples were incubated for 4 h at 4 °C on a rotating wheel with 20–100 µl of protein A magnetic beads (Dynabeads Protein A, Invitrogen, 10002D). To reduce unspecific binding, samples were washed twice in IP dilution buffer, twice in IP wash buffer 1 [100 mM Tris-HCl, pH 9.0, 500 mM LiCl, 1% (v/v) NP-40, 1% (w/v) deoxycholic acid] and once in IP wash buffer 2 [100 mM Tris-HCl pH 9.0, 500 mM LiCl, 150 mM NaCl, 1% (v/v) NP-40, 1% (w/v) deoxycholic acid]. Chromatin was eluted from the beads by incubating with 100 µl ChIP elution buffer [50 mM NaHCO₃, 1% (w/v) SDS] for 30 min at room temperature and 1400 rpm. Eluates plus 2% input sample (for H3K4me3- and H3K36me2-ChIP-seq; for XPC- and CTD-ChIP-seq, IPs from XPC⁻/⁻ cells and from empty vector-transfected ASH1L⁻/⁻ cells, respectively, served as controls) were supplemented with 1 µl proteinase K and 12.5 µl proteinase K mix at 50 °C for 3 h and at 65 °C overnight, incubated with 1 µl RNase for 20 min at 37 °C, and purified using the QIAquick PCR Purification Kit. To remove UV-induced DNA lesions with the potential to inhibit sequencing, samples were incubated with PreCR Repair mix (New England BioLabs, M0309) for 15 min at 37 °C, then re-purified, quantified by Qubit (Invitrogen, Q33238) and sent to the Functional Genomics Center Zurich for final library preparation with the NEBNext Ultra Kit (New England BioLabs, E7645). Libraries were sequenced as 100 bp single-end reads on a NovaSeq 6000 (Illumina). For the H3K36me2-ChIP-seq libraries, an additional 5–8 million 150 bp paired-end reads per sample were sequenced so as to obtain at least 45 million total fragments per sample as recommended by the ENCODE pipeline for a histone mark with a broad peak distribution.

The resulting H3K4me3-, H3K36me2-, CTD-, and XPC-ChIP-seq reads were processed using scripts from the ENCODE ChIP-seq pipeline[71]. Software citations are provided in Supplementary Table 6. Briefly, raw reads were trimmed to remove reads shorter than 99 base pairs and crop reads exceeding 101 bp (TrimGalore v.0.6.5), then aligned (bwa mem v.0.7.17-r1188) to the human reference genome build GCA hg38 lacking chrY but including non-canonical contigs. Alignments were filtered to exclude reads with mapping quality scores below 30 (bedtools v.2.29.2), PCR and optical duplicates (Picard v.2.23.8), ENCODE-blacklisted regions, non-canonical contigs, and unmapped or secondary alignments (samtools v.1.7). The additional H3K36me2 reads were processed using the ENCODE scripts for paired-end reads, trimming initial reads to remove those shorter than 149 bp and longer than 151 bp, and were pooled with the single-end reads just before calling peaks. Filtered alignments were used to call peaks (macs2 v.2.2.7.1 with default parameters; broad peaks were called for H3K36me2 and the two transcription factors, and narrow peaks for H3K4me3) and to calculate fragment extension size using R scripts (R v.4.0.3 and 4.1.0; R package Rcpp v.1.0.8.3) from the ENCODE pipeline[72,73]. Peaks were called using a *P* value cutoff of 1e⁻²; peak calling

reproducibility was verified with idr v.2.0.4.2. Only robust peaks found in a majority of biological replicates were used in downstream analyses. Fold enrichment tracks were generated of IP signal over input control (for H3K4me3 and H3K36me2), or over IP signal from ASH1L⁻/⁻ transfected with empty vector (for CTD) or XPC⁻/⁻ cells (for XPC). Boxplots were generated in R 4.1.0 (using R packages DESeq2 v.1.32.0, ggpubr v.0.4.0, ggplot2 v.3.3.5, dplyr v.1.0.8, tidyr v.1.2.0, and rstatix v.0.7.0) of continuous (fold enrichment) H3K4me3-, XPC-, and CTD-ChIP-seq signal per genomic feature bin, with signal from the full set of replicates per condition plotted in each boxplot.

## HS damage-seq
U2OS cells and (as a control) naked genomic DNA from U2OS cells were UV-C-irradiated (20 J m⁻²) as described above. The DNA was extracted from cells using the QIAamp DNA Mini kit (Qiagen) and fragmented by sonication. A detailed protocol of the library preparation[6] is provided in the Supplementary Methods. Briefly, fragmented DNA was purified using size-selective beads to remove short DNA fragments and obtain an average length of 600 bp. The DNA fragments were ligated with adapter 1 following the instructions of NEBNext Ultra II DNA Library Prep Kit for Illumina (New England Biolabs). DNA fragments containing CPD were enriched by magnetic bead-based immunoprecipitation using an anti-CPD antibody (TDM-2, Cosmo Bio). Following elution of the enriched DNA from the beads, we used NEBNext Ultra II Q5 polymerase (New England Biolabs) to extend primers up to the damage site. Next, we used subtractive hybridization to remove non-specific DNA fragments that do not contain CPDs. The resulting DNA products were subjected to adapter 2 ligation, then indexed and amplified to the desired concentration for sequencing on an Illumina NovaSeq 6000 sequencer (Illumina, San Diego, CA, USA).

Following published protocols[5,6] a single library was obtained for each experimental condition. Sequencing reads were processed using custom scripts based on the CPDSeqer pipeline[74]. Raw reads were trimmed and adapters removed with bbduk (bbmap v.38.90) followed by alignment (bwa mem v.0.7.17-r1188) to the human reference genome build GCA hg38 lacking chrY but including non-canonical contigs. Alignments were filtered to remove PCR and optical duplicates, ENCODE-blacklisted regions, and non-canonical contigs. Only alignments with mapping quality scores of at least 20 were retained (samtools v.1.7; bedtools v.2.29.2). The dinucleotide sequences immediately upstream of each read were extracted from the opposite strand and filtered to retain only dipyrimidines. The number of captured CPDs was 36–50 million per cellular sample. In wild-type cells, we captured 50 million CPDs at 0 h (immediately after UV irradiation) and there were 36 million left by 3 h of repair. Instead in ASH1L⁻/⁻ cells, there were 46 million CPDs at 0 h and 45 million by 3 h. Raw dipyrimidine counts were binned into genomic regions of interest and transformed to obtain counts per million reads (CPM) using TMM (Trimmed Mean of M-values) from the edgeR (v.3.34.1) R package (R v.4.1.0). To control for differences due to DNA sequence[6,62,74,75], CPM from each experimental sample were divided by CPM from the corresponding naked DNA control. To verify the replicability of the HS damage-seq data, we prepared an extra library from an independent culture of U2OS cells harvested immediately after UV irradiation and sequenced this replicate library to approximately half the depth of the original library. H3K4me3 peaks were used as landmarks to compare the genome-wide distribution of CPDs between the two independent samples. For that purpose, raw CPD counts from the original, replicate and naked DNA libraries were assigned to bins as outlined in Supplementary Fig. 5d, i.e., centered around the 55,000 sites of novel ASH1L-deposited H3K4me3 peaks. The binned counts from the three libraries were TMM-transformed together. Subsequently, cellular CPM divided by CPM from the naked DNA were plotted at each bin (Supplementary Fig. 10).

## Statistics and reproducibility

Tests for significance were conducted using the Prism 9.5.0 software from GraphPad. A synopsis of statistical methods and outcomes is given in Supplementary Table 5. The genome-wide CPD profiles used in Fig. 3f–h, Fig. 4, Supplementary Fig. 5h, i, and Supplementary Fig. 6c, d were obtained from a single HS-damage-seq experiment capturing 36–50 million CPDs per condition. The robustness of the resulting CPD distributions was confirmed by an independent replication as shown in Supplementary Fig. 10. The interactome of Fig. 6a was obtained from a single mass spectrometry analysis. Out of all candidates identified in this exploratory experiment, only interactions with SPT16 were confirmed by immunoblotting (Supplementary Fig. 9).

## Reporting summary

Further information on research design is available in the Nature Portfolio Reporting Summary linked to this article.

## Data availability

The data that support this study are available from the corresponding author upon reasonable request. The genomics data generated in this study have been deposited in NCBI's Gene Expression Omnibus[76] database under GEO Series accession code GSE227009. Non-sequencing data generated in this study (including the mass spectrometry raw data) are available in the OSF database (https://doi.org/10.17605/OSF.IO/WVR9C; https://osf.io/wvr9c). Additionally, the mass spectrometry proteomics data have been deposited to the ProteomeXchange Consortium via the PRIDE (http://www.ebi.ac.uk/pride) partner repository with the dataset identifier PXD043089. The data used for graphical displays are provided in the Source Data file. Sequencing reads were aligned to the GRCh38 or hg38 build of the human genome available from NCBI (https://www.ncbi.nlm.nih.gov/assembly/?term=GCA_000001405). Source data are provided with this paper.

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

## Acknowledgements

We thank Prof. Jinchuan Hu for valuable guidance in damage-seq experiments and Dr. Vakil Takhaveev for helpful support and discussion on sequencing data. We also acknowledge the Functional Genomics Center Zurich (FGCZ) and the Genetic Diversity Centre (GDC) for providing the sequencing platform and instruments for this study. The FGCZ is also acknowledged for performing mass spectrometry. This research is funded by the Swiss National Science Foundation grants 189125 (to H.N.) and 185020 (to S.J.S.).

## Author contributions

C.M. conceived the project, performed clonogenic survivals, cell cycle analysis, plasmid cloning, DNA repair assays, ChIP-seq, DNA damage-seq, immunoprecipitations and immunofluorescence studies. R.K. contributed to plasmid cloning and immunoprecipitations as well as to part of the ChIP-seq assays. M.N.Y. planned and supervised the majority of sequencing approaches, and analyzed the ChIP-seq, DNA damage-seq and ATAC-seq data. S.D. contributed to the immunofluorescence experiments and performed the ATAC-seq experiments. S.B. performed the H3K4me3-ChIP-seq experiment. N.S.F. generated the gene deletions in U2OS cells. Y.J. coordinated the DNA damage-sequencing experiments and prepared custom sequencing libraries. S.J.S. and H.N. conceived, coordinated and supervised the project. C.M., M.N.Y. and H.N. drafted the manuscript. All authors commented and edited the manuscript.

## Competing interests

The authors declare no competing interests.
