## [Peer Review File · Nature Communications]

ASH1L-MRG15 methyltransferase deposits H3K4me3 and FACT for damage verification in nucleotide excision repairREVIEWER COMMENTS

Reviewer #1 (Remarks to the Author):

Reviewers' comments for the manuscript NCOMMS-22-36442 "ASH1L methyltransferase deposits H3K4me3 and FACT for damage verification in nucleotide excision repair"

This group previously reported in 2017 that ASH1L binds to DDB2 and is recruited to UV-induced DNA damage sites to trimethylate histone H3K4, thereby instantly loading XPC (NATURE COMMUNICATIONS | DOI: 10.1038/s 41467-017-01080-8). In this study, they demonstrated that ASH1L, whose methylation activity is activated by IMG15, is recruited to the site of DNA damage, and that the H3K4me3 mark at the damage site facilitates XPC loading by comparing H3K4me3 ChIP and HS damage seq. They also demonstrated that ASH1L promotes handover from XPC to TFIIH via association of ASH1L with the histone chaperone FACT. This is an excellent study that elucidates one of the initial recognition mechanisms of the global genome-NER in the cell. It is hoped that this work will be published after further validation in several respects.

Comments

Major points,

1. In a 2017 Nat Commun paper, the authors identified ASH1L by siRNA depletion of several mRNAs involved in methylation. Since histone H3K4me3 is increased when ASH1L in the chromatin fraction after UV irradiation is slightly increased, they are focusing on H3K4me3 modification by ASH1L. However, many papers have reported that ASH1L di-methylates H3K36 in addition to tri-methylating histone H3K4. In their paper in 2017, it seems that H3K36me3 is slightly elevated after UV irradiation, although there is no statistically significant difference. In addition, in the 2016 cancer discovery paper (Cancer Discov. 2016 Jul;6(7):770-83. doi: 10.1158/2159-8290.CD-16-0058.), ASH1L catalyzes histone H3K36me2 that is read by LEDGF to recruit MLL1 and mark H3K4me3. It should be further investigated how histone H3K36me2 is involved in their study. First they should add a panel of H3K4me3 and H3K36me2 by western blotting in Fig. 1b. In addition, H3K36me2 ChIP data should be added to the experiments performed with H3K4me3 ChIP in Fig. 3 and 4. Considering Cancer Discov's paper above, they should also add comparative data between MLL1 ChIP and HS damage seq. I want the authors to add one more experiment: can the ASH1L-dependent H3K4me3 hotspot be altered by MLL1 knockdown?

2. In this paper, they examined whether ASH1L methylation activity is required for the progression of NER using CTD and CTD inactive. It is unclear to what extent this CTD complements the physiological activity of ASH1L full length. To make this point clearer, they should describe what we know about the N-terminal domain of ASH1L, which is missing in CTD. In addition, they should compare the localization changes of ASH1L FL and CTD before and after UV irradiation by ASH1L ChIP seq and H3K4me3 ChIP seq to confirm that the complementation experiment by CTD truly complements ASH1L FL.

3. The interpretation of the data in Fig. 5 is unconvincing to the reviewer. To begin with, it is necessary to show how is the time course of accumulation of XPCs at the damaged site after UV irradiation in U2OS cells. Then, when the accumulation of XPCs is maximal in WT cells, whether it is lower or higher in ASH1L^{-/-} cells must be shown. Similar experiments in other papers have shown that if ASH1L functions upstream of XPC, it should be downregulated. The same is true for MRG15 knockdown cells. Since this paper lacks DDB2^{-/-} data, it is not possible to compare the effects of upstream factors on XPCs; data from DDB2^{-/-} or knockdown cells must be shown side by side.

4. The authors have identified SSRP1 and SPT16, which are components of the histone chaperone FACT, as molecules involved in the turnover of XPC to TFIIH by mass spectrometry. It is this part that can be said to be a new discovery in this paper, and I would like to see a more detailed analysis. First, SSRP1 and SPT16 were included in the immunoprecipitates of CTD, MRG15 and XPC, respectively, but

were they due to direct binding or were they identified via chromatin at damaged sites? It should be clarified whether if it is an indirect binding, I think they should show data or at least discuss how FACT is recruited to the CPD site. Does FACT recognize histone modification and perform chaperone functions? If it is direct, it should be examined which region of CTD, MRG15 or XPC molecules binds to SSRP1 or SPT16.

Minor points,

5. Their UDS assay is not common and it is not certain how they quantified the incorporated EdU. Also, although the amounts of DNA to be applied are aligned, the DNA content of the samples with quantified CPD amounts is not corrected by PI or DAPI. I would like to know if their assay method and the general method (UV irradiation of whole cells to incorporate EdU and quantification of DNA by PI or DAPI) produce the same results.

6. It is necessary to put a title for the X-axis of the graphs in Fig. 3h and Fig. 4.

7. The authors should explain how the zeros on the x-axis in Fig. 4d-f were determined.

8. Recently, one paper has been published, which reported that MLL1 catalyzed histone H3K4me3 to recruit chromatin remodeler BAZ1A and facilitate GG-NER (Biochim Biophys Acta Mol Cell Res. 2022 Nov;1869(11):119332. doi: 10.1016/j.bbamcr.2022.119332.). They should discuss whether the histone H3K4me3 modification by ASH1L promotes recruitment of BAZ1A, and whether MLL1 and ASH1L may work together or separately.

Reviewer #2 (Remarks to the Author):

In the manuscript "ASH1L methyltransferase deposits H3K4me3 and FACT for damage verification in nucleotide excision repair", Maritz et.al showed that the activated histone methyltransferase ASH1L could accelerate global genome NER activity. They provide evidence to show that ASH1L deposits H3K4me3 marks all over the genome (except in gene promoters), thus priming chromatin for relocations of XPC from native to damaged DNA. Histone chaperone FACT can be recruited to UV lesions via binding to ASH1L. In the absence of ASH1L, MRG15 or FACT, XPC persists on damaged DNA without being able to deliver lesions to the TFIIH verifier. They concluded that ASH1L mediated ectopic H3K4me3 and FACT occupancy confers an active promoter-like code and organization of histones that make DNA damage verifiable by the NER machinery. Overall, the study appears to be interesting but the enthusiasm was dampened by substantial concerns. One of the major concerns is the overall lack of detailed molecular mechanism on how ASH1L/ H3K4me3 recruits NER proteins to CPDs, The other concern involves the lack of physiological relevance. The specific comments are below:

1. Although authors provide biochemical evidence that ASH1L may play a role in NER, however, the physiological relevance of ASH1L was not evaluated in this study, for example, if ASH1L knockout sensitize cells to UV exposure?
2. Figure 1, Authors claimed that ASH1L sharply accelerates the GG-NER reaction, although they showed that ASH1L depletion impaired GG-NER reaction, experiments involving ectopic ASH1L is required to support the conclusion that ASH1L sharply accelerates the GG-NER reaction.
3. Figure 2C-2E, a full-length ASH1L construct is lacking, which again raise the concern that if ectopic ASH1L promotes the GG-NER reaction?
4. Figure 3, the author should show that if XPC globally co-localize with H3K4me3 with or without UV exposure?
5. Figure 3, one of the concern is that H3K4me3 signal mainly enriched in euchromatin, but how it can influence the NER efficacy in heterochromatin, which is marked by H3K9me3?
6. Figure 3, is possible, an ASH1L-ChIP-seq should be performed, and analyze the co-localization of

ASH1L and XPC or TFIIH, as it can help to further elucidate that the enzymatic activity but not the scaffolding function is required for the effect on GG-NER reaction.

7. Figure 4, the authors found that CPD excision takes place mainly at de novo H3K4me3 marks, in general, H3K4me3 mainly marks promoter, and sometimes includes enhancer; In addition, unlike H3K9me3, the H3K4me3 peak is narrow and sharp. In a word, H3K4me3 peaks cover very limited genomic region. Given that CPDs spread across the genome, how those CPDs beyond H3K4me3 could be repaired?

8. Since UV exposure result in numerous de novo H3K4me3 marks, the author should investigate the location of those H3K4me3 peaks and correlates with RNAseq data; for example, if the de novo H3K4me3 modify the enhancers or super-enhancers, which promotes specific gene expression to facilitate the NER process?

9. The authors showed that ASH1L used its methyltransferase function to accelerate GG-NER (Figure 2), and H3K4me3 appears to be essential for XPC relocation, but the detailed mechanism is still not clear, for example, does XPC recognize H3K4me3 and directly bind to it?

10. In Figure 6, the authors showed that ASH1L recruits FACT for the XPC-to-TFIIH handover. Co-ip or pull-down by using full-length ASH1L or CTD constructs should be performed to verify the ASH1L and FACT subunits protein interaction.

11. Figure 6K, CTDinactive seems did not influence the SPT16 recruitment to CPD spots, while they also showed that the methyltransferase function of ASH1L is essential for GG-NER stimulation (Figure 2), then how to explain these contradictory observations?

Reviewer #3 (Remarks to the Author):

Nucleotide excision repair is the major mechanism for removal of bulky damages from the human genome. NER is divided into two sub-pathways, transcription-coupled repair (TC-NER) and global genome repair (GG-NER). GG-NER efficiency is highly influenced by chromatin accessibility. With time, the vast majority of damages are eventually repaired even in heterochromatic region, but it is still unclear how repair occurs in nucleosomal templates. In this paper, Maritz et al. provide compelling evidence for the involvement of ASH1L and H3K4me3 post-translational modification in facilitating this repair.

The authors previous Nat. Comm publication in 2017 showed that ASH1 is recruited to chromatin after UV by DDB2, and that it is necessary for efficient GG-NER and UV survival. ASH1L mutations result in lower CPD removal and lower repair synthesis – but do not affect recovery of RNA synthesis (TC-NER). They also showed that XPC (through a specific binding domain) is recruited to H3K4me3.

In this papers, the authors extended on their previous findings. They generated ASH1L knock-out cells and showed that like the previous siRNA treated cells they are UV sensitive and have impaired CPD repair. Here, by complementing knock out cells with wt or catalytically dead ASH1L they provide more mechanistic insight showing the repair defect is dependent on the methyltransferase activity. Furthermore, depletion of MRG15 – a regulator necessary for the methyltransferase activity resulted in impaired NER. In response to UV, DDB1 recruits MRG15 to activate NER.

To further understand the function of ASH1L in NER across the genome, they performed ChIP-seq for H3K4me3 and for XPC in response to UV. Interestingly, they found UV-induced H3K4me3 induced ChIP-seq peaks that were dependent on ASH1L. They relate the XPC localization to the H3K4 me3 distribution and deduce that XPC relocates to sites of H3K4me3.

Following genome-wide repair with Damage-seq, the authors show that the novel H3K4me3 sites are

sites of relatively better repair compared to UV-induced but ASH1L independent H3K4me3 or random sites.

In the absence of ASH1L XPC persist on the lesion, and the recruitment of the TFIIH subunit XPD is impaired. This recruitment is again dependent on the methyltransferase activity of ASH1L. Using IP and mass spec the authors found that the FACT complex components SPT16 and SSRP1 bind the CTD of ASH1L and MRG15. Knocking down SPT16 and SSRP1 resulted in deficient NER and XPD recruitment. Recruitment of SPT16 was not dependent on the methyltransferase activity.

Together, this led the authors to propose a clear and elegant model that in response to UV, ASH1L is recruited to damage sites by DDB2 and methylates H3K4 outside of promoters, by that facilitating recruitment of XPC and initiation of GG-NER in nucleosomal templates. ASH1L recruits the FACT complex which then promotes the turnover of XPC and recruitment of TFIIH for repair.

Major comments.

1) The authors had a previous publication in Nature Comm. identifying ASH1L as a major regulator of GG-NER. The language claiming that this manuscript reports ASH1L as a novel regulator could be toned down.

2) Regarding the redistribution of H3K4me3 after UV.

a. The results the authors show for novel peaks that are ASH1L are persuasive. Still, since UV damages form relatively randomly across the genome, we would not expect, and from Figure 3b/c – do not see strong (high signal to noise) peaks of H3K4me3. It is important to add statistical analysis (significance values) to the panels in Figure 3 – especially 3b.

b. H3K4me3 is a mark of active promoters. After UV – there is a process of transcriptional shut down that could in itself reduce the levels of H3K4me3 at promoters and explain some of the redistribution of H3K4me3 even in the absence of ASH1L. This should be mentioned/discussed.

c. The authors show in Figure 4a that the novel ASH1L-dependent, H3K4me3 peaks – have higher CPD damage levels. CPDs are reported to have hot-spots at transcription factor binding sites (ETS1 specifically, PMIDs: 29980679, 31723047, 30586386) and are generally dependent on the underlying TT frequency (33817640). Are the ASH1L dependent peaks especially rich in TTs or do they have specific TF binding motifs?

3) Regarding the XPC redistribution.

The XPC ChIP-seq is problematic and it's difficult from the figures to ascertain if it indeed worked or whether we are simply looking at noise from a failed ChIP. This is an inherent challenge with ChIP for repair proteins recruited to random genomic sites. The authors state they performed experiments in XPC^{-/-} cells as control and perhaps data should be more clearly compared (for example in Figure 3d). Furthermore statistics must be presented. The differences in XPC occupancy in Figure 3e is of a maximum of ~6% and this is very small given the possible noise between replicates in the ChIP-seq protocol. Did the authors look at the XPC profile over the H3K4me3 sites as they did for damage and repair in Figure 4?

* I would like to note that if this ChIP-seq data is eventually omitted or moved to supplement the manuscript would still be compelling.

4) Regarding CPD damage and repair at H3K4me3 sites.

a. Regarding all the CPD Damage-seq data: There is no mention of number of replicates in the manuscript and I could not find out since it does not appear in the GEO accession provided.

b. The authors show convincing evidence that CPD repair is elevated at the H3K4me3 sites induced by UV in an ASH1L dependent manner. On page 13 the authors state "after UV irradiation, CPD excision is largely confined to the locations of these H3K4me3 marks (Figs. 3 and 4)." - however this was not shown – and could be misleading. CPD repair is elevated at accessible sites across the genome (PMID: 27036006) - that were not analyzed here. For instance- the authors did not compare repair at the UV-induced peaks to repair at promoter/non UV-induced H3K4me3 sites – which were previously associated with higher repair. The fact that regions that were accessible before UV are repaired efficiently does not diminish from the importance of having regions that gain accessibility to allow repair.

4) Analysis of genome accessibility (by ATAC-seq for example) at the H3K4me3 sites after UV could strengthen the model (especially if it were SPT16 dependent!).

The results the authors present are thorough and persuasive and constitute an important advance in our understanding of NER in chromatin. If these comments are adequately addressed this manuscript will be very suitable for Nature communications.

We are very grateful to all three reviewers for their very valuable comments and criticisms. Please find here our point-by-point responses:

Reviewer #1:

Major points,

1. In a 2017 Nat Commun paper, the authors identified ASH1L by siRNA depletion of several mRNAs involved in methylation. Since histone H3K4me3 is increased when ASH1L in the chromatin fraction after UV irradiation is slightly increased, they are focusing on H3K4me3 modification by ASH1L. However, many papers have reported that ASH1L di-methylates H3K36 in addition to tri-methylating histone H3K4. In their paper in 2017, it seems that H3K36me3 is slightly elevated after UV irradiation, although there is no statistically significant difference. In addition, in the 2016 cancer discovery paper (Cancer Discov. 2016 Jul;6(7):770-83. doi: 10.1158/2159-8290.CD-16-0058.), ASH1L catalyzes histone H3K36me2 that is read by LEDGF to recruit MLL1 and mark H3K4me3. It should be further investigated how histone H3K36me2 is involved in their study. First they should add a panel of H3K4me3 and H3K36me2 by western blotting in Fig. 1b. In addition, H3K36me2 ChIP data should be added to the experiments performed with H3K4me3 ChIP in Fig. 3 and 4. Considering Cancer Discov's paper above, they should also add comparative data between MLL1 ChIP and HS damage seq. I want the authors to add one more experiment: can the ASH1L-dependent H3K4me3 hotspot be altered by MLL1 knockdown?

Response: The reviewer would like to know whether, besides H3K4me3, H3K36me2 deposited by ASH1L, possibly resulting in the recruitment of MLL1, is required for the GG-NER reaction. However, our **new Figs. 2j** and **2k** show that, unlike the down regulation of ASH1L, a knockdown of MLL1 has minimal, not statistically significant effects on GG-NER activity (determined by three biologically completely independent experiments; significance test by one-way ANOVA). Notably, the siRNA-mediated depletion of MLL1 protein was highly efficient (**new Suppl. Fig. S3i**) and the repair assay was accompanied by appropriate controls (siControl, siASH1L and siXPA). In view of this finding that MLL1 is not or only marginally involved in the GG-NER reaction, we did not carry out further experiments with this particular methyltransferase and instead focused on ASH1L. Conversely, we established genome-wide H3K36me2-ChIP-seq tracks as requested by the reviewer, but found that, unlike H3K4me3 peaks, H3K36me2 peaks arising after UV irradiation are deposited away from CPD sites, and that these novel H3K36me2 peaks do not correlate with preferential CPD excision (**new Suppl. Fig. S6c**). We actually found poor CPD repair in the center of these H3K36me2 peaks, leading to the conclusion that H3K36me2 marks do not stimulate GG-NER activity. These new findings are described in the revised manuscript (p. 4, lines 16-20 and p. 6, lines 35-46).

2. In this paper, they examined whether ASH1L methylation activity is required for the progression of NER using CTD and CTD inactive. It is unclear to what extent this CTD complements the physiological activity of ASH1L full length. To make this point clearer, they should describe what we know about the N-terminal domain of ASH1L, which is missing in CTD. In addition, they should compare the localization changes of ASH1L FL and CTD before and after UV irradiation by ASH1L ChIP seq and H3K4me3 ChIP seq to confirm that the complementation experiment by CTD truly complements ASH1L FL.

Response: In the revised manuscript, we added information on the N-terminal portion of ASH1L, pointing out that no function as yet has been assigned to this part of the protein, except that it contains “AT-hooks”, which are putative DNA-binding motifs (p. 3, lines 25-26). It is unfortunately not possible to conduct ASH1L-ChIP experiments because no suitable antibody exists to pull down ASH1L. However, we followed the reviewer’s recommendation and determined genome-wide ChIP-sequencing tracks of the carboxy-terminal domain (CTD), which contains the catalytic site and all known protein interaction motifs. These additional genome-wide tracks demonstrate that the CTD of ASH1L is recruited to chromatin in response to UV irradiation (**new Suppl. Fig. S5c**), that this recruitment of CTD occurs preferentially at CPD sites (**new Suppl. Fig. S6d**, panel on the left), that CTD co-localizes with sites of novel ASH1L-deposited H3K4me3 peaks (**new Fig. 3d**) and that this preferred localization of CTD in response to UV irradiation translates to enhanced excision (**new Suppl. Fig. S6d**, panel on the right). Thus, these new results strongly support our conclusion that ASH1L stimulates GG-NER activity by the deposition of H3K4me3 marks at lesion sites (p. 5 of the revised manuscript, lines 4-11, and p. 6, lines 42-46).

3. The interpretation of the data in Fig. 5 is unconvincing to the reviewer. To begin with, it is necessary to show how is the time course of accumulation of XPCs at the damaged site after UV irradiation in U2OS cells. Then, when the accumulation of XPCs is maximal in WT cells, whether it is lower or higher in ASH1L^{-/-} cells must be shown. Similar experiments in other papers have shown that if ASH1L functions upstream of XPC, it should be downregulated. The same is true for MRG15 knockdown cells. Since this paper lacks DDB2^{-/-} data, it is not possible to compare the effects of upstream factors on XPCs; data from DDB2^{-/-} or knockdown cells must be shown side by side.

Response: Following the reviewer’s recommendation, we provide in **Fig. 5a** a time course of XPC recruitment to the spots of UV irradiation. The finding remains unchanged: the lack of ASH1L in ASH1L^{-/-} cells does not affect the overall recruitment of XPC to the UV spots 1 h after irradiation. Instead, the lack of ASH1L in ASH1L^{-/-} cells causes an abnormal retention of XPC at the 3-h time point after irradiation compared to wildtype controls (**Figs. 5a** and **5b**). Exactly the same is observed with the MRG15 knockdown (**Figs. 5c** and **5d**). Upon the reviewer’s request, we also included in the revised manuscript experiments with DDB2 depletion, clearly demonstrating that, as expected, the recruitment of XPC to UV lesion spots at the 3-h time point is fully dependent on DDB2 (**Suppl. Figs. S7a** and **S7b**; see also the revised manuscript text on p. 7, lines 14-17).

4. The authors have identified SSRP1 and SPT16, which are components of the histone chaperone FACT, as molecules involved in the turnover of XPC to TFIIH by mass spectrometry. It is this part that can be said to be a new discovery in this paper, and I would like to see a more detailed analysis. First, SSRP1 and SPT16 were included in the immunoprecipitates of CTD, MRG15 and XPC, respectively, but were they due to direct binding or were they identified via chromatin at damaged sites? It should be clarified whether If it is an indirect binding, I think they should show data or at least discuss how FACT is recruited to the CPD site. Does FACT recognize histone modification and perform chaperone functions? If it is direct, it should be examined which region of CTD, MRG15 or XPC molecules binds to SSRP1 or SPT16.

Response: To address these reviewer’s comments, we have included in the revised manuscript additional immunoprecipitation studies that confirm by Western blotting the mass spectrometry finding that SPT16 co-exists in solubilized and chromatin-free protein

complexes with MRG15 and ASH1L (see p. 8 of the revised manuscript, lines 29-34, and **new Suppl. Fig. S9**). Briefly, Flag-tagged MRG15 and the Flag-tagged CTD of ASH1L were expressed in HEK293T cells. Lyzed cells were treated with benzonase to completely digest chromatin. Anti-Flag antibodies were used for immunoprecipitations, resulting in both cases in the co-immunoprecipitation of SPT16. As indicated in the mechanistic model of **Fig. 7**, these findings imply that the ASH1L-MRG15 complex provides a docking site for the recruitment of FACT to damaged chromatin. This conclusion is supported by the observation that the recruitment of SPT16 (one of two FACT subunits) to UV lesion spots is nearly abolished in ASH1L^{-/-} cells (**Figs. 6g** and **6h**) or by the depletion of MRG15 or ASH1L (**new Fig. 6i**). In terms of specific protein regions of ASH1L, we concluded that the CTD portion is sufficient to mediate these interactions for the SPT16 recruitment. We also generated constructs coding for smaller portions (SET domain with and without the BAH domain), but U2OS and HEK cells unfortunately failed to express them.

Minor points,

5. Their UDS assay is not common and it is not certain how they quantified the incorporated EdU. Also, although the amounts of DNA to be applied are aligned, the DNA content of the samples with quantified CPD amounts is not corrected by PI or DAPI. I would like to know if their assay method and the general method (UV irradiation of whole cells to incorporate EdU and quantification of DNA by PI or DAPI) produce the same results.

Response: The measurement of repair synthesis in local spots of UV damage is widely used in the field and we have included the new Reference no. 69 as an example of recent report, in which this exact method has been applied (revised manuscript p. 11, lines 35-36). The normalization is performed by subtracting the unspecific fluorescence background in the nucleus from the specific fluorescence signal within the UV lesion spots (see p. 12 of the revised manuscript, lines 1-3). We do not favor the method proposed by the reviewer (UV irradiation of whole cells and normalization by quantification of DNA) because this approach makes it more difficult to correct for nascent DNA replication synthesis that would not be fully normalized by measuring the overall DNA content. Nevertheless, we repeated the experiment of **Fig. 1f** using the reviewer's method and found the same pronounced NER defect in ASH1L^{-/-} cells compared to wildtype. Notably, the results of this NER assay are confirmed by two independent methods, i.e., the immunoassay of **Fig. 1c** and the HS damage-seq assay of **Fig. 3f**.

6. It is necessary to put a title for the X-axis of the graphs in Fig. 3h and Fig. 4.

Response: The x-axes of **Fig. 3h** and **Fig. 4** are now appropriately labeled with "Distance from center of H3K4me3 peaks", "Distance from center of novel ASH1L-deposited H3K4me3 peaks" and "Distance from center of random regions".

7. The authors should explain how the zeros on the x-axis in Fig. 4d-f were determined.

Response: The zero values on the x-axis of **Fig. 4** are explained in the **new Suppl. Fig. S5d** and its corresponding figure legend. Briefly, the centers of methylation peaks were assigned to a bin or DNA segment of 64 base pairs. These center bins were expanded out by increments of 64 bp, thus covering up to ± 2 kilobases away from the peak centers. Sequencing reads derived from CTD-ChIP-seq (**new Fig. 3d**), XPC-ChIP-seq (**new Fig. 3e**) or HS damage-seq (**Figs. 4a** and **4c**) were mapped to each bin in this region of ± 2 kilobases.

Similarly, CPD excision rates (during the first 3 h after the UV pulse) were plotted around the centers of histone methylation (**Fig. 3h, Fig. 4b**). In the case of the control panels of **Figs. 4d-f**, the zero values correspond to the center of randomly selected 654-base pair regions not overlapping the novel ASH1L-deposited methylation peaks (where 654 bp matches the average size of these peaks). This is now explained on p. 22, lines 37-38 of the revised manuscript.

8. Recently, one paper has been published, which reported that MLL1 catalyzed histone H3K4me3 to recruit chromatin remodeler BAZ1A and facilitate GG-NER (Biochim Biophys Acta Mol Cell Res. 2022 Nov;1869(11):119332. doi: 10.1016/j.bbamcr.2022.119332.). They should discuss whether the histone H3K4me3 modification by ASH1L promotes recruitment of BAZ1A, and whether MLL1 and ASH1L may work together or separately.

Response: As already detailed above, our **new Figs. 2j and 2k** show that, unlike the down regulation of ASH1L, a depletion of MLL1 has minimal, statistically insignificant effects on GG-NER activity (determined with three completely independent experiments; significance test by one-way ANOVA). Notably, the siRNA-mediated depletion of MLL1 protein was highly efficient (**new Suppl. Fig. S3i**) and the repair assay was accompanied by appropriate controls (siControl, siASH1L and siXPA). In view of this finding that MLL1 is not or only marginally involved in the GG-NER reaction, we did not carry out further experiments with this particular methyltransferase and instead focused on ASH1L. However, the paper mentioned by the reviewer is now included in the reference list (Reference number 45) and both the BAZ1A chromatin remodeler and MLL1 are discussed in the main text (see bottom of p. 9 and top of p. 10 of the revised manuscript).

Reviewer #2 (Remarks to the Author):

Overall, the study appears to be interesting but the enthusiasm was dampened by substantial concerns. One of the major concerns is the overall lack of detailed molecular mechanism on how ASH1L/ H3K4me3 recruits NER proteins to CPDs, The other concern involves the lack of physiological relevance.”

Response: With the revised manuscript we now present evidence for a complete chain of events starting with the recruitment of ASH1L to UV lesions (using the carboxy-terminal domain of ASH1L as a surrogate; **new Fig. 3d**), the deposition by ASH1L of H3K4me3 marks (**Figs. 3a-c**), the recruitment of XPC to the H3K4me3 marks (**new Fig. 3e**), the recruitment of FACT (**Fig. 6**), the recruitment of TFIIH (**Fig. 5**), finally culminating in the preferential excision of CPDs at the H3K4me3-marked sites (**Figs. 3f-h and Fig. 4**). The molecular mechanism by which ASH1L induces GG-NER activity is summarized in **Fig. 7**. The physiologic relevance is indicated by the increased sensitivity of ASH1L^{-/-} cells to UV light (see response to comment no. 1 below).

The specific comments are below:

1. Although authors provide biochemical evidence that ASH1L may play a role in NER, however, the physiological relevance of ASH1L was not evaluated in this study, for example, if ASH1L knockout sensitize cells to UV exposure?

Response: **Suppl. Figs. S2a and S2b** show that the *ASH1L* deletion sensitizes the cells to UV exposure. **Suppl. Figs. 2c-e** show that the *ASH1L* deletion also impacts on the cell cycle upon UV irradiation (with a reduced population of S-phase cells and an increased fraction of cells in G2/M, see p. 2, lines 37-40 of the revised manuscript).

2. Figure 1, Authors claimed that ASH1L sharply accelerates the GG-NER reaction, although they showed that ASH1L depletion impaired GG-NER reaction, experiments involving ectopic ASH1L is required to support the conclusion that ASH1L sharply accelerates the GG-NER reaction.

Response: We found in rescue experiments that expression of the carboxy-terminal domain (CTD) of ASH1L is sufficient to rescue all the described effects of the ASH1L deletion, including impaired GG-NER activity (**Figs. 2c-e**), delayed turnover of XPC at lesion sites (**Figs. 5e and 5f**), and the impaired recruitment of TFIIH (**Figs. 5k and 5l**) and of FACT (**Figs. 6j and 6k**). As highlighted in the revised manuscript (p. 3, lines 21-26), this carboxy-terminal domain of >100 kDa contains the catalytic site and all known protein interaction motifs and, also in view of its proven functionality in the GG-NER pathway, can be considered a valid surrogate of the full-length counterpart. Experiments with full-length ASH1L are unfortunately not possible because of its low ectopic expression and the lack of effective antibodies for immunofluorescence, IP and ChIP-seq experiments.

3. Figure 2C-2E, a full-length ASH1L construct is lacking, which again raise the concern that if ectopic ASH1L promotes the GG-NER reaction?

Response: Rescue assays and other experiments with full-length ASH1L are unfortunately not possible because of its low ectopic expression and the lack of effective antibodies for immunofluorescence, IP and ChIP-seq experiments. However, the carboxy-terminal domain (CTD) of ASH1L is fully functional in the GG-NER reaction (**Figs. 2c-e**) and, as described above, is able to fully complement all GG-NER defects. This carboxy-terminal domain has been also successfully used for genome-wide ChIP-seq assays (**new Fig. 3d and new Suppl. Fig. S5c**). Thus, the carboxy-terminal domain provides a valid surrogate of the full-length protein for the purpose of our study focusing on its GG-NER function.

4. Figure 3, the author should show that if XPC globally co-localize with H3K4me3 with or without UV exposure?

Response: The **new Fig. 3e**, requested by the reviewer, demonstrates that the repartitioning of XPC upon UV irradiation indeed results in its global colocalization with H3K4me3 peaks that are newly deposited by ASH1L. This strong enrichment of XPC at sites of novel ASH1L-dependent H3K4me3 peaks is missing in unchallenged cells.

5. Figure 3, one of the concern is that H3K4me3 signal mainly enriched in euchromatin, but how it can influence the NER efficacy in heterochromatin, which is marked by H3K9me3?

Response: The revised manuscript contains new ATAC-seq (assay for transposase accessible chromatin with sequencing) data (**new Suppl. Fig. S8**), demonstrating that UV exposure, presumably through the action of chromatin remodelers, leads to an increased chromatin accessibility throughout the genome including at inactive intergenic regions (“heterochromatin”), which is a prerequisite for repair activity. According to our ATAC-seq

data, however, ASH1L does not influence this increase of accessibility (p. 8, lines 9-16 of the revised manuscript). As outlined in the legend to **Fig. 7**, the role of ASH1L is, instead, to act as a caliper to gauge whether chromatin is amenable to downstream NER factors, primarily the TFIIH complex for lesion verification. Our report implies that the successful deposition of H3K4me3 at lesion sites by ASH1L provides a signal for the recruitment of XPC to lesions that are sufficiently accessible to be processed by the large GG-NER complex.

6. Figure 3, is possible, an ASH1L-ChIP-seq should be performed, and analyze the co-localization of ASH1L and XPC or TFIIH, as it can help to further elucidate that the enzymatic activity but not the scaffolding function is required for the effect on GG-NER reaction.

Response: We established CTD-ChIP-seq tracks (using the CTD as a functional surrogate of full-length ASH1L, see response to comment No. 3 above). The data show a genome-wide co-localization of CTD with the sites of ASH1L-deposited H3K4me3 peaks (p. 5, lines 4-11, and **new Fig. 3d**) and also a genome-wide co-localization of XPC with the same ASH1L-deposited H3K4me3 peaks (**new Fig. 3e**). Taken together, these findings further confirm that the enzymatic activity of ASH1L mediates its GG-NER role, as already shown by the rescue experiments with CTD_{active} and CTD_{inactive} of **Figs. 2c-e**. In addition, these experiments prove the sequential co-localization of CTD and XPC at sites of novel ASH1L-deposited H3K4me3 peaks: the CTD of ASH1L deposits the H3K4me3 marks, which are subsequently recognized by XPC to trigger the GG-NER reaction.

7. Figure 4, the authors found that CPD excision takes place mainly at de novo H3K4me3 marks, in general, H3K4me3 mainly marks promoter, and sometimes includes enhancer; In addition, unlike H3K9me3, the H3K4me3 peak is narrow and sharp. In a word, H3K4me3 peaks cover very limited genomic region. Given that CPDs spread across the genome, how those CPDs beyond H3K4me3 could be repaired?

Response: Our data show that new H3K4me3 peaks are deposited at CPD sites throughout the genome (**Figs. 3b and 3c, Fig. 4a**). Thus, the GG-NER system does not have to rely on preexisting H3K4me3 marks. Instead, novel H3K4me3 marks are deposited in response to UV irradiation at sites of CPD formation across the entire genome to stimulate GG-NER activity.

8. Since UV exposure result in numerous de novo H3K4me3 marks, the author should investigate the location of those H3K4me3 peaks and correlates with RNAseq data; for example, if the de novo H3K4me3 modify the enhancers or super-enhancers, which promotes specific gene expression to facilitate the NER process?

Response: The overall distribution of the *de novo* ASH1L-dependent H3K4me3 peaks in response to UV irradiation is shown in **Fig. 3b**. With respect to gene promoters, 100 of 251 p53 stress response targets (<https://doi.org/10.1016/j.celrep.2014.06.030>), whose transcription presumably increases upon UV irradiation, overlap with such novel ASH1L-deposited H3K4me3 peaks. With respect to enhancer elements, we found that 11% of U2OS enhancers from the EnhancerAtlas 2.0 database overlap with novel ASH1L-deposited H3K4me3 peaks. Conversely, 12% of the novel ASH1L-deposited H3K4me3 peaks are added to the enhancers. To avoid an impact of transcriptional effects on NER activity, in the present study we focused on repair incubations of no more than 3-4 h after UV exposure. During this short time after the UV challenge, in wildtype cells we did not observe any induction of the p53-dependent

XPC and DDB2 proteins that could account for the drastic reduction of initial repair kinetics in *ASH1L*-deleted or *ASH1L*-depleted cells compared to controls. In particular, the unchanged levels of XPC protein up to 3-4 h after UV irradiation discouraged us from performing transcriptional analyses during this early time period and an expansion to later time points (for example 24 h post UV) is beyond the scope of the present manuscript.

9. The authors showed that *ASH1L* used its methyltransferase function to accelerate GG-NER (Figure 2), and H3K4me3 appears to be essential for XPC relocation, but the detailed mechanism is still not clear, for example, does XPC recognize H3K4me3 and directly bind to it?

Response: We previously reported that recombinant XPC protein interacts with recombinant H3K4me3, and we identified a H3K4me3-binding site in XPC protein, i.e., a short β -turn motif located between its two well-characterized β -hairpin domains BHD2 and BHD3 (<https://doi.org/10.1038/s41467-017-01080-8>). This information is now included in the revised manuscript (see p. 24, lines 2-4).

10. In Figure 6, the authors showed that *ASH1L* recruits FACT for the XPC-to-TFIIH handover. Co-ip or pull-down by using full-length *ASH1L* or CTD constructs should be performed to verify the *ASH1L* and FACT subunits protein interaction.

Response: As requested by the reviewer, we have included in the revised manuscript additional co-IP studies that confirm by Western blotting the mass spectrometry finding that SPT16 co-exists in soluble protein complexes with MRG15 and *ASH1L* (see p. 8 of the revised manuscript, lines 29-34 and **new Suppl. Fig. S9**). Briefly, Flag-tagged MRG15 and Flag-tagged CTD were expressed in HEK293T cells. After benzonase treatment to completely digest nucleic acids, solubilized cell lysates were subjected to immunoprecipitations. Anti-Flag antibodies were used for immunoprecipitations, resulting in both cases in the co-immunoprecipitation of SPT16 (one of the FACT subunits).

11. Figure 6K, CTDinactive seems did not influence the SPT16 recruitment to CPD spots, while they also showed that the methyltransferase function of *ASH1L* is essential for GG-NER stimulation (Figure 2), then how to explain these contradictory observations?

Response: The revised manuscript contains a more detailed model that better explains the seemingly “contradictory” observation as pointed out by the reviewer (see new legend to Figure 7 on p. 23 and p. 24). Briefly, *ASH1L* exploits a bimodal mechanism to make DNA damage verifiable. First, *ASH1L* primes nucleosomes for XPC binding by the deposition of H3K4me3, thus navigating this GG-NER initiator away from constitutive locations on native DNA towards damaged sites. The methyltransferase function, culminating in the deposition of H3K4me3, is indispensable for this step and acts as a caliper to gauge whether chromatin is amenable to large downstream NER factors, primarily the TFIIH complex. Second, *ASH1L* itself uses H3K4me3 as an anchor to remain associated with the methylated nucleosome, thus providing a docking site for the recruitment of the histone chaperone FACT. At this step, a non-enzymatic matchmaker function is sufficient to couple FACT to the GG-NER machinery. This model accounts for the observation that, in the absence of the enzymatic activity of *ASH1L*, XPC still detects DNA damage but fails to be escorted to chromatin sites that are amenable to the follow-up damage verification by TFIIH. Because of

this misplacement, XPC persists at lesion sites and despite the presence of FACT is unable to deliver the DNA substrate to the TFIID complex.

Reviewer #3 (Remarks to the Author):

Major comments.

1) The authors had a previous publication in Nature Comm. identifying ASH1L as a major regulator of GG-NER. The language claiming that this manuscript reports ASH1L as a novel regulator could be toned down.

Response: The abstract, the last paragraph of the introduction and the discussion section have been rephrased following the reviewer's recommendation. We now make clear throughout these sections that the present manuscript describes the molecular process by which ASH1L stimulates GG-NER activity (see for example p. 2, line 15 of the revised manuscript).

2) Regarding the redistribution of H3K4me3 after UV.

a. The results the authors show for novel peaks that are ASH1L are persuasive. Still, since UV damages form relatively randomly across the genome, we would not expect, and from Figure 3b/c – do not see strong (high signal to noise) peaks of H3K4me3. It is important to add statistical analysis (significance values) to the panels in Figure 3 – especially 3b.

Response: We have added standard deviations to **Figs. 3a** and **3b**. Significance values were added to **Fig. 3b** upon application of one-way ANOVA as the statistical test. The p-values for the changes described in **Fig. 3b** range from < 0.01 to < 0.0001 , supporting the conclusion that ASH1L deposits extra H3K4me3 marks in response to DNA damage induced by UV irradiation. Detailed results of statistical tests have been compiled in **Suppl. Table S5**.

b. H3K4me3 is a mark of active promoters. After UV – there is a process of transcriptional shut down that could in itself reduce the levels of H3K4me3 at promoters and explain some of the redistribution of H3K4me3 even in the absence of ASH1L. This should be mentioned/discussed.

Response: We do not see a reduction of H3K4me3 peaks in active promoters upon UV irradiation, as the density of these peaks remains unchanged in active promoters. On the other hand, there is a statistically significant increase of H3K4me3 peak frequency in inactive (including “poised”) promoters after the UV challenge. Therefore, we cannot confirm the reviewer's hypothesis that there is a “redistribution” of H3K4me3 from promoters to other genomic features. In the absence of ASH1L, there were no significant changes in H3K4me3 peak density across the different genomic features (**Fig. 3b**).

c. The authors show in Figure 4a that the novel ASH1L-dependent, H3K4me3 peaks – have higher CPD damage levels. CPDs are reported to have hot-spots at transcription factor binding sites (ETS1 specifically, PMIDs: 29980679, 31723047, 30586386) and are generally dependent on the underlying TT frequency (33817640). Are the ASH1L dependent peaks especially rich in TTs or do they have specific TF binding motifs?

Response: We apologize for having caused confusion about the sequence of events (CPD formation → H3K4me3 deposition, not H3K4me3 → CPD). **Fig. 4a** demonstrates that H3K4me3 peaks are deposited preferentially at CPD sites after the formation of these lesions upon UV irradiation. The revised text now reads: "...there was at the level of the entire genome a correlation between the initial CPD formation upon UV exposure and the follow-up addition of the ~55,000 H3K4me3 peaks by ASH1L (**Fig. 4a**)" (p. 6, lines 26-28). The corresponding revised figure legend reads as follows: "In the first 3 h after UV irradiation, ASH1L deposits H3K4me3 peaks (determined by H3K4me3-ChIP-seq) mainly at sites of CPD induction (determined by HS damage-seq). The ~55,000 genomic sites harboring *de novo* ASH1L-dependent H3K4me3 peaks deposited post UV and their flanking sequences were subdivided into 64-bp bins as illustrated in **Supplementary Fig. S5d**. The mean initial abundance of CPDs is indicated for each of these bins" (p. 22, lines 25-30). In any case, we examined the data for di-pyrimidine or transcription factor binding motif frequency and found that of the 16 possible dinucleotide combinations comprising the novel H3K4me3 peaks, TT ranked only 7th most frequently within the peaks (and the 3 other possible di-pyrimidine combinations ranked 3rd, 5th, and 9th); furthermore, pre-existing H3K4me3 peaks contained three- to twelve-fold higher density of ETS1 binding sites (from five non-U2OS cell types; ENCODE Project Consortium) than did the novel ASH1L-dependent peaks, further supporting the sequence of events described above (CPD formation followed by H3K4me3 deposition).

3) Regarding the XPC redistribution.

The XPC ChIP-seq is problematic and it's difficult from the figures to ascertain if it indeed worked or whether we are simply looking at noise from a failed ChIP. This is an inherent challenge with ChIP for repair proteins recruited to random genomic sites. The authors state they performed experiments in XPC^{-/-} cells as control and perhaps data should be more clearly compared (for example in Figure 3d). Furthermore statistics must be presented. The differences in XPC occupancy in Figure 3e is of a maximum of ~6% and this is very small given the possible noise between replicates in the ChIP-seq protocol. Did the authors look at the XPC profile over the H3K4me3 sites as they did for damage and repair in Figure 4?

*** I would like to note that if this ChIP-seq data is eventually omitted or moved to supplement the manuscript would still be compelling.**

Response: We agree with the reviewer that the analysis of XPC-ChIP-seq data is challenging in view of the prominence of this factor in chromatin. However, following the reviewers' recommendation, we were able to determine XPC profiles at novel ASH1L-dependent peaks. The **new Fig. 3e** now unequivocally demonstrates that a fraction of XPC moves to these novel ASH1L-dependent H3K4me3 peaks in response to UV damage (p. 5, lines 26-28 of the revised manuscript). We also followed the reviewers' recommendation by moving the data on the overall distributions of XPC into the supplementary section (**new Suppl. Figs. S5e-g**).

4) Regarding CPD damage and repair at H3K4me3 sites.

a. Regarding all the CPD Damage-seq data: There is no mention of number of replicates in the manuscript and I could not find out since it does not appear in the GEO accession provided.

Response: In the revised manuscript, we make clear that, following the protocol of Hu and coworkers (laboratory of Dr. A. Sancar, Ref. no. 6), a single library (with millions of sequencing reads per library) was prepared for each experimental condition and libraries were

re-sequenced to obtain sufficient coverage (p. 15, lines 8-9). The number of captured CPDs was 36-50 million per cellular sample: in wildtype cells, we captured 50 million CPDs at 0 h (immediately after UV exposure) and there were 36 million left by 3 h; in ASH1L^{-/-} cells, there were 46 million CPDs at 0 h and 45 million by 3 h (p. 15, lines 17-20). In view of the variability of the resulting CPD excision rates across the genome, the differences between wildtype and ASH1L^{-/-} cells (**Fig. 3f**) and between different types of H3K4me3 peaks (**Fig. 3g**) were subjected to Wilcoxon rank sum tests for statistical significance.

b. The authors show convincing evidence that CPD repair is elevated at the H3K4me3 sites induced by UV in an ASH1L dependent manner. On page 13 the authors state “after UV irradiation, CPD excision is largely confined to the locations of these H3K4me3 marks (Figs. 3 and 4).” - however this was not shown – and could be misleading. CPD repair is elevated at accessible sites across the genome (PMID: 27036006) - that were not analyzed here. For instance- the authors did not compare repair at the UV-induced peaks to repair at promoter/non UV-induced H3K4me3 sites – which were previously associated with higher repair. The fact that regions that were accessible before UV are repaired efficiently does not diminish from the importance of having regions that gain accessibility to allow repair.

Response: To address this concern, we added more CPD excision data. First, we normalized the data of **Fig. 3f** for the different amounts, depicted on the figure inset, of genomic DNA in each chromatin feature. These normalized CPD excision rates demonstrate that, as expected, the overall CPD repair is more efficient in promoters than for example in heterochromatin (**new Suppl. Fig. 5i**). Second, the revised **Fig. 3h** compares the repair rates at novel ASH1L-dependent H3K4me3 peaks in intergenic regions (left panel) with the repair rates at ASH1L-independent H3K4me3 peaks detected across the entire genome (middle panel) and the repair rates at pre-existing H3K4me3 peaks, again across the entire genome (right panel). These new data suggest that the H3K4me3 peaks are preferential sites of CPD excision. This view is confirmed in **Fig. 4** covering the full genome, where we show that CPD repair takes place preferentially at novel ASH1L-deposited H3K4me3 peaks (**Fig. 4b**) compared to random sequences across the entire genome (**Fig. 4e**). We believe that these results support the conclusion that CPD lesions are preferentially excised at novel ASH1L-deposited H3K4me3 peaks. The sentence from the discussion quoted by the reviewer has been changed to “Consequently, after UV irradiation, CPD excision occurs preferentially at the locations of these novel ASH1L-deposited H3K4me3 marks (Figs. 3 and 4)” (p. 9, lines 37-39).

4) Analysis of genome accessibility (by ATAC-seq for example) at the H3K4me3 sites after UV could strengthen the model (especially if it were SPT16 dependent!).

Response: We included in the revised manuscript a thorough analysis of genome accessibility by ATAC-seq (**new Suppl. Fig. S8**; see new text on p. 8, lines 9-16). We agree with the reviewer that one hypothesis was that ASH1L (through recruitment of SPT16) induces chromatin relaxation that favors DNA accessibility. Indeed, the ATAC-seq analysis demonstrated an increased accessibility upon UV irradiation compared to unchallenged cells, particularly in gene bodies and intergenic regions. However, this improved access to DNA was indistinguishable between wildtype and ASH1L^{-/-} cells, indicating that the ASH1L-MRG15 complex promotes the XPC-to-TFIIH transition by another mechanism. As outlined in the legend to **Fig. 7**, rather than promoting accessibility, ASH1L probes the existing accessibility of DNA lesions across the genome and, by depositing H3K4me3, attaches a flag to those damaged sites that are amenable to the recruitment of large NER complexes,

including particularly the TFIID complex.

REVIEWER COMMENTS

Reviewer #1 (Remarks to the Author):

The authors have addressed all of the reviewer's questions logically and without contradiction. I believe that this paper is worthy of acceptance because it contains content appropriate for nature communications.

Reviewer #2 (Remarks to the Author):

The authors performed additional experiments and revised manuscript, while the quality of paper improved; however, several major concerns were not completely addressed:

1. For the response to point 2 of this reviewer, the author explained that full-length ASH1L are not possible because of its low ectopic expression and lack of effective antibodies for immunofluorescence, ChIP-seq experiments, however, the expression level may vary in different cell lines, the author should try more cell lines, because the carboxy-terminal domain (CTD) of ASH1L is actually not present in wild types cell lines, the expression of full-length ASH1L and test its effect on NER proteins recruitment to CPDs would be important for support the conclusion that "ASH1L sharply accelerates the GG-NER reaction" by recruiting NER proteins to CPDs.

2. For the response to point 5 of this reviewer, although the authors demonstrated that UV exposure leads to increased chromatin accessibility throughout the genome including at heterochromatin, they still need to compare the NER repair efficiency in the region with or without the H3K4me3, because it seems not possible that all CPD sites are marked by de novo H3K4me3, especially for those CPD sites locate in the heterochromatin.

Reviewer #3 (Remarks to the Author):

The analysis of repair by UDS clearly showed the importance of ASH1L for CPD removal, so the damage-mapping results do not "stand-alone", and are supportive of the authors findings. Still, I have some concern that the damage mapping experiments were performed with a single replicate.

Aside from this, I find the manuscript has been improved in revision and I am satisfied with how my comments were addressed. It is an impressive and important contribution to our understanding of repair in chromatin and I recommend it for publication in Nature Communications.

Please find here our detailed responses to the remaining concerns from the reviewers:

Reviewer #2

1. For the response to point 2 of this reviewer, the author explained that full-length ASH1L are not possible because of its low ectopic expression and lack of effective antibodies for immunofluorescence, ChIP-seq experiments, however, the expression level may vary in different cell lines, the author should try more cell lines, because the carboxy-terminal domain (CTD) of ASH1L is actually not present in wild types cell lines, the expression of full-length ASH1L and test its effect on NER proteins recruitment to CPDs would be important for support the conclusion that “ASH1L sharply accelerates the GG-NER reaction” by recruiting NER proteins to CPDs.

Response: We agree that the expression level may vary in different cell lines; therefore, we previously transfected HeLa, U2OS, HEK293T and CHO cells with a vector coding for appropriately tagged full-length ASH1L. In all cases, however, we were unable to detect any ectopic ASH1L protein by Western blot. We also did not find any antibody that would allow for immunofluorescence and immunoprecipitation studies with endogenous ASH1L in HeLa, U2OS or HEK293T cells. The decision to work with the 110 kDa-carboxy-terminal domain, which contains the catalytic site and all known protein interaction motifs, was based on its ability to fully complement the DNA repair defect of ASH1L^{-/-} cells (**Fig. 2**). In our view, this decision is justified by the outcome of the mechanistic studies carried out with this carboxy-terminal domain using ChIP-seq, immunoprecipitation and immunofluorescence methods (**Figs. 3, 5 and 6; Supplementary Fig. S9**).

2. For the response to point 5 of this reviewer, although the authors demonstrated that UV exposure leads to increased chromatin accessibility throughout the genome including at heterochromatin, they still need to compare the NER repair efficiency in the region with or without the H3K4me3, because it seems not possible that all CPD sites are marked by de novo H3K4me3, especially for those CPD sites locate in the heterochromatin.

Response: The reviewer is right when stating that “...it seems not possible that all CPD sites are marked by de novo H3K4me3.” Our manuscript indeed contains the following quantitative assessment: “Within the first 3 h after irradiation, ~55,000 such novel H3K4me3 peaks were deposited across the genome of wildtype cells, translating to at least 1 extra peak every 10 CPDs or every 500 nucleosomes” (p. 4, lines 185-187). This rather low ratio of H3K4me3 peaks to CPDs is consistent with the slow rate of CPD excision with half-lives of around 24 hours reported in the literature (see also the CPD excision assay of **Fig. 1c**).

However, we do show that CPD excision is most efficient at ASH1L-deposited H3K4me3 peaks. We initially assessed CPD excision in intergenic regions of the genome, where we expected low transcriptional activity and, hence, little overlap with transcription-coupled repair. **Fig. 3g** provides a direct quantitative comparison of CPD excision efficiency at three different features of intergenic regions: high CPD excision from sites of ASH1L-deposited H3K4me3 peaks (feature No. 1), low CPD excision from H3K4me3 peaks that are independent of ASH1L (feature No. 2) and essentially no CPD excision from random sites outside H3K4me3 peaks (feature No. 3). Also, in **Fig. 3h**, we directly compared CPD excision efficiencies at these ASH1L-deposited H3K4me3 peaks of intergenic regions (left

panel), at ASH1L-independent H3K4me3 peaks detected across the entire genome (middle panel) and at pre-existing H3K4me3 peaks, again across the entire genome (right panel). These data show that the ASH1L-deposited H3K4me3 peaks are preferential sites of CPD excision. This view is further confirmed in **Fig. 4** covering the full genome, where we show that CPD excision takes place preferentially at ASH1L-deposited H3K4me3 peaks (**Fig. 4b**) compared to the same number of random sites (**Fig. 4e**). We conclude that, as illustrated in **Fig. 7**, ASH1L acts as a caliper to gauge whether chromatin is amenable to downstream NER factors, primarily TFIIH. By the deposition of H3K4me3, ASH1L provides a signal for the recruitment of XPC to lesions that are sufficiently accessible to be processed by the global-genome NER complex.

Reviewer #3

The analysis of repair by UDS clearly showed the importance of ASH1L for CPD removal, so the damage-mapping results do not "stand-alone", and are supportive of the authors findings. Still, I have some concern that the damage mapping experiments were performed with a single replicate.

Response: We thank the reviewer for sharing this concern. With the HS damage-seq experiment, we could reach sufficient coverage for reliable genome-wide damage mapping capturing 50 million CPDs after UV radiation. The reproducibility of the resulting CPD profiles was tested by a biologically independent replicate (p. 15, lines 697-706 of the revised manuscript). In both the original and replicate experiment, UV-irradiated U2OS cells were subjected to HS damage-seq analyses. The sites of ASH1L-deposited H3K4me3 peaks were then used as landmarks to compare genome-wide damage distributions. As shown in the new **Supplementary Fig. S10**, the CPD maps obtained in the main HS damage-seq experiment and in its replicate were essentially superimposable, thus demonstrating the robustness of the HS damage-seq mapping procedure. To summarize, we present three completely independent methods demonstrating the global-genome NER defect of ASH1L^{-/-} cells: the excision assay of **Fig. 1c**, the UDS assay of **Figs. 1 e-g** and its complementation (**Figs. 2c-e**) and, finally, the HS damage-seq experiment of **Figs. 3f-h**.

REVIEWERS' COMMENTS

Reviewer #2 (Remarks to the Author):

The author basically addressed my concerns, and the expression of full-length ASH1L can be done later. I recommend it for publication in Nature Communications.

Reviewer #3 (Remarks to the Author):

I am satisfied with the revised version of the manuscript and recommend it for publication in Nature Communications.